# Reprogramming mechanism dissection and trophoblast replacement application in monkey somatic cell nuclear transfer

Zhaodi Liao [1,2,3,5], Jixiang Zhang [3,4,5], Shiyu Sun [1,2,3,5], Yuzhuo Li[1,2], Yuting Xu[1,2], Chunyang Li[1,2], Jing Cao[1,2], Yanhong Nie[1,2], Zhuoyue Niu[3,4], Jingwen Liu[3,4], Falong Lu [3,4,6] ✉, Zhen Liu [1,2,6] ✉ & Qiang Sun [1,2,6] ✉

Somatic cell nuclear transfer (SCNT) successfully clones cynomolgus monkeys, but the efficiency remains low due to a limited understanding of the reprogramming mechanism. Notably, no rhesus monkey has been cloned through SCNT so far. Our study conducts a comparative analysis of multi-omics datasets, comparing embryos resulting from intracytoplasmic sperm injection (ICSI) with those from SCNT. Our findings reveal a widespread decrease in DNA methylation and the loss of imprinting in maternally imprinted genes within SCNT monkey blastocysts. This loss of imprinting persists in SCNT embryos cultured in-vitro until E17 and in full-term SCNT placentas. Additionally, histological examination of SCNT placentas shows noticeable hyperplasia and calcification. To address these defects, we develop a trophoblast replacement method, ultimately leading to the successful cloning of a healthy male rhesus monkey. These discoveries provide valuable insights into the reprogramming mechanism of monkey SCNT and introduce a promising strategy for primate cloning.

Somatic cell nuclear transfer (SCNT) technology has been widely utilized for the cloning of various mammalian species, including sheep[1], cattle[2], mouse[3], swine[4], goat[5], rabbit[6], and dog[7]. In recent advancements, wild-type and gene-edited cynomolgus monkeys (*Macaca fascicularis*) have been successfully cloned using SCNT. This achievement has been made possible through the optimization of the SCNT protocol and the application of epigenetic regulators to overcome the epigenetic challenges during pre-implantation. Notably, the inclusion of histone demethylase *Kdm4d* and the histone deacetylation inhibitor trichostatin A (TSA) has resulted in the live birth rates of 2.5% and 1.5%[8,9], respectively. The successful cloning of rhesus (*Macaca mulatta*) monkeys was first reported back in 1997, utilizing blastomeres from early-stage embryos as donor cells[10]. However, it is worth noting that

until recently, there had no postnatally viable somatic cell cloned rhesus monkey. Although a recent report claimed the birth of a somatic cell cloned rhesus monkey that survived for less than 12 h after birth[11].

In conventional cloning methods, the live birth rates for most mammalian species are extremely low, ranging from 1% to 3%, with slightly higher rates observed for bovines (5%–20%)[12,13]. Consistent with the low survival rates, developmental anomalies, particularly in the extraembryonic lineages[12], have been reported in somatic cell cloned animals,. For example, SCNT piglets have exhibited lower villi density and thickness in the extravillous cytotrophoblast layer of the placenta compared to artificial insemination-derived ones[14]. Bovine and sheep SCNT embryos have shown fetal losses due to defects in

[1]Institute of Neuroscience, Center for Excellence in Brain Science and Intelligence Technology, State Key Laboratory of Neuroscience, Chinese Academy of Sciences, Shanghai 200031, China. [2]Shanghai Center for Brain Science and Brain-Inspired Technology, Shanghai 201210, China. [3]University of Chinese Academy of Sciences, Beijing 100049, China. [4]State Key Laboratory of Molecular Developmental Biology, Institute of Genetics and Developmental Biology, Chinese Academy of Sciences, Beijing 100101, China. [5]These authors contributed equally: Zhaodi Liao, Jixiang Zhang, Shiyu Sun. [6]These authors jointly supervised this work: Falong Lu, Zhen Liu, Qiang Sun. ✉e-mail: fllu@genetics.ac.cn; zliu2010@ion.ac.cn; qsun@ion.ac.cn

placentome formation, volume density, and placenta structure[15,16]. Cloned mice have also displayed abnormal distribution of the spongiotrophoblast and labyrinthine layers in enlarged SCNT placentas[17,18]. Furthermore, aberrant expression of imprinted genes has been observed in the placentas of cloned mice and trophoblast stem cells derived from SCNT[17,19–21]. Although there is limited research on pathological studies in cloned non-human primates (NHPs), a very recent study reported functional deficiencies in several visceral organs due to uterine hypoxia in cloned rhesus monkeys[11]. Previous studies have explored the viability of the trophoblast exchange technique to achieve live births[22], but its effectiveness in improving the cloning efficiency of bovines has been limited[23]. Notably, defects in the trophoblast cell lineage have been found to impact the development of cloned mice[24].

The successful long-term survival of somatic cell cloned rhesus monkeys is of great importance, as these NHPs are extensively used for basic and clinical research[10,11,25–27]. Despite previous attempts using the same protocol as for cloning of cynomolgus monkeys[8,9], achieving this goal has remained elusive. In this study, we aimed to improve the cloning efficiency of rhesus monkeys by employing a strategy known as trophoblast replacement (TR), also referred to as SCNT-TR, in combination with treatment using the epigenetic regulators *Kdm4d* and TSA. For the first time, we established a TR method specifically for rhesus monkey by injecting the inner cell mass (ICM) derived from SCNT embryos into the blastocoeles (with the ICM removed) derived from intracytoplasmic sperm injection (ICSI) embryos. Remarkably, using this approach, we successfully achieved the live birth of a healthy SCNT rhesus monkey that has survived for over 2 years at the time of preparing this research for publication. We performed Sanger sequencing and site-specific deep sequencing on the ectoderm and mesoderm derivatives of this SCNT-TR monkey and found no evidence of chimerism contributed by the ICM of the trophoblast donor embryo.

## Results

### Development failure of the post-implantation SCNT embryos

Similar to the approach used for cynomolgus monkey cloning, we employed a combination of TSA and *Kdm4d* treatment for rhesus monkey cloning. We observed that the rhesus SCNT embryos showed normal chromosome organization in their first M-phase (Supplementary Fig. 1). Consistently, the blastocyst rate of rhesus monkey SCNT embryos reached 47.6% (10/21), which is slightly lower than that of the ICSI group but comparable to our previous report on cynomolgus cloning (Fig. 1a)[8,9]. With such a high blastocyst rate, we transferred a total of 484 cloned rhesus embryos into 96 surrogates, with an average of 5 embryos per surrogate (Table 1). However, we observed that the implantation rate of the cloned rhesus embryos was less than half that of the ICSI rhesus embryos (74/499 vs 35/484, Fig. 1b). Additionally, in comparison to the live birth rate of fetuses derived from normal fertilization, most of the SCNT fetuses were lost during the post-implant stage (Fig. 1c and Table 1). Although one out of the 35 implanted SCNT fetuses was live-born, it did not survive for longer than 23 h postnatally (Table 1).

To investigate the stage at which most of the cloned rhesus fetuses were lost, we analyzed their survival probability during each 10-day window of the gestation period. It was observed that the highest abortion rate occurred around day 150 for ICSI embryos, whereas most of the cloned rhesus embryos were lost around day 40 of gestation (Fig. 1d). This finding was further supported by the density plot, which indicated that the peak of abortion for both cloned and ICSI rhesus fetuses were around day 40 and day 140 of the gestation period (Fig. 1e). To confirm if this pattern was consistent in cynomolgus monkeys as well, data from both cynomolgus and rhesus monkeys were included. After analyzing a total of 213 and 166 aborted fetuses in the ICSI and SCNT groups, respectively, it

was discovered that approximately half of the cloned fetuses were lost around day 60 of gestation, while half of the ICSI fetuses were lost around day 130 of the gestation period (Fig. 1f). The density plot of the abortion rate displayed a notable contrast between the SCNT and ICSI groups (Fig. 1g).

Overall, these findings suggest that the reprogramming defects in cloned NHP embryos manifest during the implantation or periimplantation stages.

### Global demethylation of the cloned monkey embryos

To gain insights into the somatic cell reprogramming of NHPs, we performed RNA-seq and whole genome (WGS) bisulfite sequencing (WGBS) analyses on ICSI and SCNT blastocysts (Fig. 2a). In maximize the number of single nucleotide polymorphisms (SNPs), the SCNT and ICSI blastocysts were generated using the hybridization of the cynomolgus and rhesus monkeys (Fig. 2a and Supplementary Fig. 2a–c). Additionally, we incorporated WGBS datasets of monkey oocytes, hybrid monkey somatic cells, and human sperm[28] for comparison. To facilitate allele-specific analysis, we sequenced the WGS of the parental individuals (Supplementary Table 1).

We observed a significantly lower DNA methylation level in SCNT blastocysts compared to ICSI blastocysts (30.0% versus 39.6%, Fig. 2b and Supplementary Fig. 2d). Furthermore, pairwise comparisons of DNA methylation in blastocysts, donor cells, and Sp +Oo (the average of sperm and oocyte) confirmed the finding (Fig. 2c). Through comprehensive genome-wide scanning analysis, we identified 19,089 differentially methylated regions (DMRs), with each DMR encompassing a minimum of 25 CpGs and exhibiting at least 25 significantly differentially methylated CpGs. Among these DMRs, the majority (10,076) displayed lower DNA methylation levels in SCNT blastocysts compared with that of ICSI blastocysts, and thus termed as hypoDMRs (Fig. 2d and Supplementary Fig. 2e). Conversely, the remaining 9013 regions exhibited higher DNA methylation levels in SCNT blastocysts compared to ICSI blastocysts and were designated as hyperDMRs (Fig. 2d and Supplementary Fig. 2f). Notably, hypoDMRs were found to have a relatively shorter average length (5542 bp) when compared to hyperDMRs (12,309 bp) (Fig. 2e). Additionally, hyperDMRs were primarily located in intergenic regions, while hypoDMRs were relatively enriched in both intergenic and intron regions, differing from hyperDMRs (Supplementary Fig. 2g).

We next proceed to investigate the mechanisms underlying the generation of these DMRs. Interestingly, we observed that the methylation levels at hyperDMRs were higher compared to the flanking regions in fibroblasts, but significantly lower than the flanking regions in the sperm and oocyte (Fig. 3a). Importantly, despite the global reduction in DNA methylation levels in both ICSI and SCNT blastocysts, hyperDMRs persisted from the somatic cell to the blastocyst stage in SCNT embryos (Fig. 3a). Further analysis revealed that the DNA methylation levels of both parental genomes in ICSI blastocyst were lower than that of SCNT embryos (Fig. 3b).

In the case of hypoDMRs, we found that the DNA methylation levels at these regions were lower than the flanking regions in both sperm (~75% versus ~80%) and oocyte (~60% versus 65%), with the difference being more pronounced in monkey fibroblasts (Fig. 3c). Interestingly, in blastocysts, the DNA methylation levels of hypoDMRs in ICSI embryos were comparable to the flanking regions, while remaining lower in SCNT embryos (Fig. 3c). Considering the global demethylation of both gametic and somatic DNA methylation during embryo development, and the fact that both SCNT and ICSI embryos undergo the same rounds of replicationdependent dilution, it is likely that the hypoDMRs were inherited from the fibroblasts. Moreover, by splitting the parental genomes of both the ICSI and SCNT blastocysts, we found that the maternally-

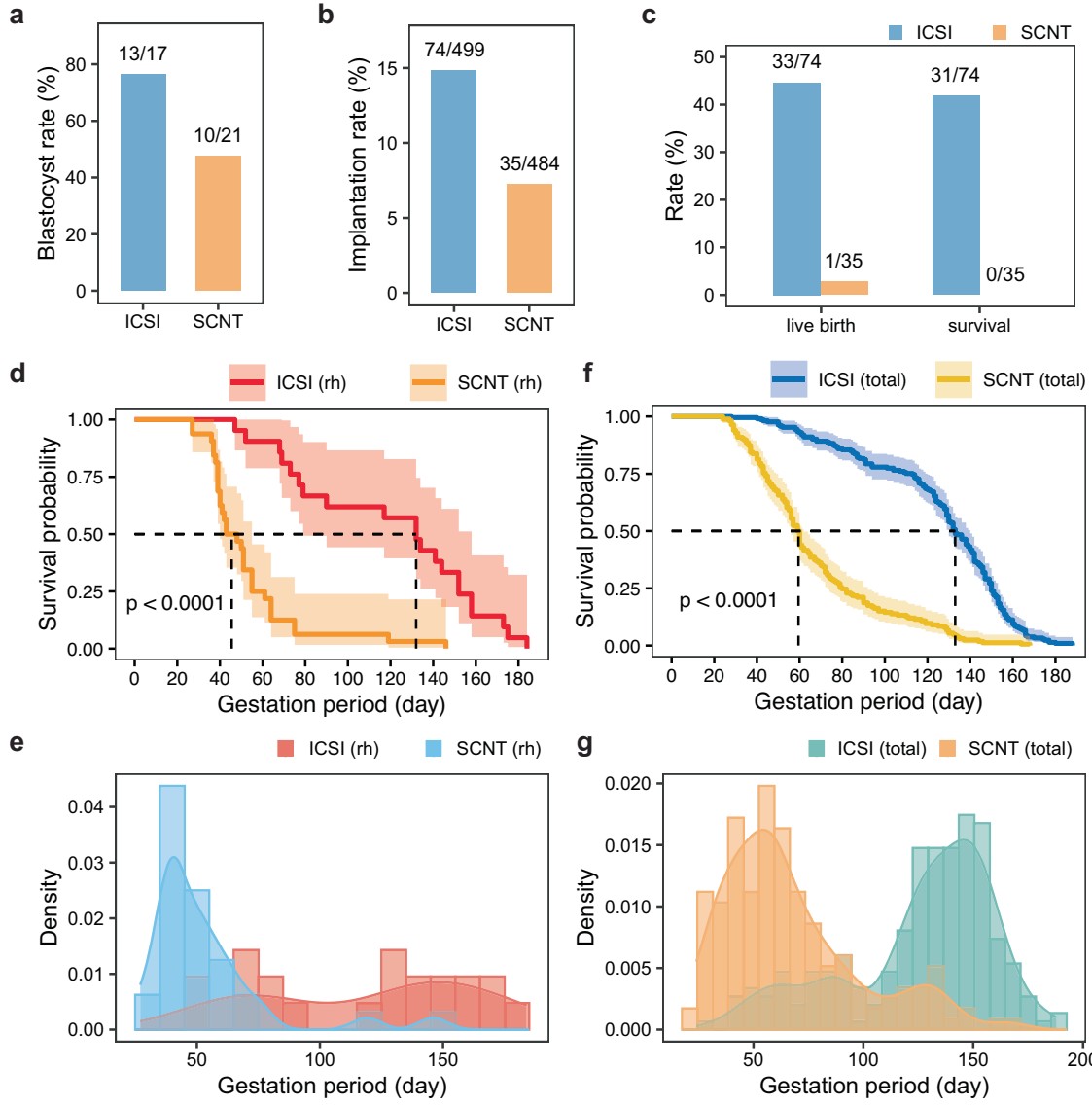

**Fig. 1 | The rapid loss of cloned monkey embryos following implantation. a** The blastocyst rate of the rhesus embryos generated by ICSI and SCNT. **b** The implantation rate of the rhesus ICSI and SCNT embryos. **c** The live birth rate and postnatal survival rate of rhesus fetuses derived from ICSI and SCNT. **d** The survival curves illustrate the rates of abortion or stillbirth in rhesus fetuses within the ICSI and SCNT groups at different gestational stages. The ICSI group comprises 21 aborted fetuses (*n* = 21), while the SCNT group includes 32 aborted fetuses (*n* = 32). The dashed lines represent the median survival times, which indicate the gestational period at which half of the fetuses experienced abortion. The survival curves were generated using the "ggsurvplot" function in R, and the *p*-value (<0.0001) was computed using the default log-rank test within the "ggsurvplot" function. **e** Density plot illustrating the occurrence of abortion or stillbirth in rhesus ICSI and SCNT fetuses at different gestation period. There are 21 aborted fetuses (*n* = 21) for

the ICSI group, and 32 aborted fetuses (*n* = 32) for the SCNT group. **f** The survival curves display the abortion or stillbirth rates in rhesus and cynomolgus monkey fetuses (ICSI and SCNT) during the post-implantation stage. The ICSI group consists of 213 aborted fetuses (*n* = 213), while the SCNT group comprises 166 aborted fetuses (*n* = 166). The dashed lines represent the median survival times, signifying the gestation period at which half of the fetuses experienced abortion. These survival curves were generated using the "ggsurvplot" function in R, and the *p*-value (<0.0001) was computed using the default log-rank test within the "ggsurvplot" function. **g** Density plot demonstrating the peak occurrence of abortion in rhesus and cynomolgus (also called total) monkey fetuses (ICSI and SCNT) at different gestation periods. There are 213 aborted fetuses (*n* = 213) for the ICSI group, and 166 aborted fetuses (*n* = 166) for the SCNT group.

biased DNA methylation in the hypoDMRs of normal fertilized blastocysts was abnormally lost in the cloned blastocysts (Fig. 3d). This loss of maternally-biased DNA modification was also observed throughout the WGS after performing parental genome splitting (Fig. 3e, f).

To summarize, these results indicated that while ICSI embryos undergo separate and ordered reprogramming of DNA methylation in the parental genomes, somatic DNA methylation modifications may persist until the blastocyst stage in SCNT embryos, even after global DNA demethylation. This ultimately leads to the generation of a large number of hypo- and hyperDMRs in SCNT blastocysts.

## Loss of the maternal imprinting in the cloned embryos

Imprinting genes play critical role during embryo development[29, 30], as their disruption can result in embryonic disability or lethality both pre- and postnatally[31,32]. Therefore, we conducted a comparison of the gene expression and DNA methylation levels of imprinting genes between SCNT and ICSI blastocysts. Due to limited research on genomic imprinting in NHPs, we utilized candidate imprinting genes from the human database[30] (https://www.geneimprint.com/site/genes-by-species). We identified 115 out of 471 human imprinting genes that showed homologous expression in our monkey blastocyst RNA-seq data (Fig. 4a and Supplementary Fig. 3a, b).

**Table 1 | Post-implant development of rhesus SCNT and SCNT-TR embryos**

| | Total embryos activated | No. of blastocysts used | No. of embryos transferred | No. of surrogates | No. of pregnant surrogates | Implanted embryos (%)[a] | Fetuses (%)[a] | Live births (%)[†] | Survival infants (%)[a] |
|---|---|---|---|---|---|---|---|---|---|
| SCNT | 484 | / | 484 | 96 | 22 | 35 (7.2) | 32 (6.6) | 1 (0.2) | 0 (0) |
| SCNT-TR[b] | 113 | 32 | 11 | 7 | 2 | 3 (2.7) | 3 (2.7) | 1 (0.9) | 1 (0.9) |

[a]Data were calculated based on the number of embryos activated for SCNT;
[b]Details of the SCNT-TR embryos reconstructed can be seen in the Supplementary Table 6.

We initially selected 33 candidate imprinting genes with fragments per kilobase of exon model per million mapped fragments (FPKM) > 1 (Fig. 4a, b). Subsequently, we mapped these candidate genes to the hyperDMRs and hypoDMRs and identified 15 out of 33 genes that were associated with DMRs (Fig. 4c, Supplementary Table 2). Using a stringent criterion (fold change >2 or fold change <0.5), we eventually obtained 3, namely *THAP3*, *DNMT1* and *SIAH1*, whose aberrant expression was correlated with the loss of DNA methylation in monkey SCNT embryos (Fig. 4c). Although we were unable to identify any SNPs in these genes between their parental genomes of ICSI and SCNT embryos from whole-genome bisulfite sequencing (WGBS) dataset (Supplementary Fig. 4a–c), we confirmed that *THAP3* lost its paternal-specific gene expression and became biallelically expressed in the SCNT embryos using parental specific SNP information (Fig. 4d and Supplementary Fig. 4d).

Next, we filtered candidate aberrantly imprinted genes from the pool of differentially expressed genes (DEGs) and found 8 differentially expressed imprinting genes out of 420 DEGs between SCNT and ICSI embryos (*p*-value < 0.01, fold change >2, baseMean >10, Fig. 4e, f). Among these, three genes (*DNMT1*, *RHOBTB3* and *THAP3*) were associated with the loss of DNA methylation in SCNT blastocysts (Fig. 4g). We confirmed the loss of maternally biased DNA methylation in *RHOBTB3* by performing allelic analysis using the WGBS data (Fig. 4h and Supplementary Fig. 4e). Additionally, the paternal-specific gene expression of *RHOBTB3* in ICSI blastocyst was lost in SCNT embryos (Fig. 4i). One thing to be noted that the SNPs we identified at the *RHOBTB3* locus between the parental genomes of ICSI embryos reside in the intron regions rather than the exon regions. This makes us unable to accurately quantify the parental specific expression of *RHOBTB3* in ICSI embryos (Supplementary Fig. 4f).

We also found that these identified genes (*THAP3*, *DNMT1*, *RHOBTB3* and *SIAH1*) lost DNA methylation in their corresponding DMRs of both the fibroblast and SCNT blastocysts (Fig. 4h and Supplementary Fig. 4a–c). This result suggests that DNA methylation from some of the imprinted genes are lost what is then inherited from somatic cells to SCNT blastocysts. On the other hand, other genes such as *CDKN1C*, *KLHDC10*, *ZFAT*, *ADTRP* and *ATP10A* were either upregulated or downregulated in monkey SCNT blastocysts compared to that of ICSI blastocysts (Fig. 4b, f), but these genes were not correlated with DMRs.

To further analyze the expression of imprinted genes in extraembryonic tissues, we analyzed trophoblasts from monkey ICSI and SCNT blastocysts (Supplementary Table S1). Consistent with observations in the whole blastocysts, we observed a significantly lower global DNA methylation level in SCNT trophoblasts (SCNT-tro) compared to ICSI trophoblasts (ICSI-tro, Supplementary Fig. 5a). Subsequent, we identified hyperDMRs and hypoDMRs between ICSI-tro and SCNT-tro (Supplementary Fig. 5b). Through parental genome allelic analysis, we observed the loss of maternal-biased DNA methylation in SCNT-tro (Supplementary Fig. 5d). Next, we examined the allelic gene expression of genes seen loss of imprinting in blastocysts, in trophoblast cells. We observed an obvious loss of the paternal-specific expression of *THAP3* in trophoblasts derived from SCNT embryos which is supported by the detection of ample SNPs between the parental genomes (Supplementary Fig. 5e). *SIAH1* exhibited clear biallelic expression in the

trophoblasts of SCNT embryos, although no single SNPs were identified between the parental genomes of *SIAH1* in the RNA-seq dataset of ICSI trophoblasts (Supplementary Fig. 5f).

Overall, we identified a total of 4 aberrantly imprinted genes in the SCNT blastocyst, namely *THAP3*, *DNMT1*, *SIAH1* and *RHOBTB3*. Among them, *DNMT1* and *THAP3* were confirmed as DEGs with FPKM > 1. These findings suggest that genomic imprinting is lost for several imprinted genes in SCNT embryos, which may play a critical role in the development of monkey SCNT embryos. It is not clear however how many other genes with imprinting defects were present in the SCNT blastocysts as the number of imprinted genes that we were able to examine was limited due to the requirement of using SNP information in our analysis pipeline.

### Continuously loss of imprinting in post-implanted monkey SCNT embryos

To investigate the DNA methylation state of these genes in the postimplanted embryos, we cultured the post-implanted monkey ICSI and SCNT embryos using a recently published method[33]. On day 17 after fertilization (E17), we collected the cultured post-implanted embryos and conducted WGBS analysis (Supplementary Fig. 6). In order to obtain as many ICM lineages as possible, we isolated the epiblast (E), amnion (A), yolk sac (Y), and a small number of trophoblast residues (T, or E/A/Y/T) in the center of the embryos using a micropipette under a microscope (Fig. 5a and Supplementary Fig. 6a).

Surprisingly, contrary to our observation of a global lower of DNA methylation in SCNT blastocysts compared to ICSI blastocysts, we found that the DNA methylation level of the E/A/Y/T tissues in cloned embryos was globally higher than that of the ICSI embryos (Fig. 5b, c left). However, the four genes (*DNMT1*, *THAP3*, *SIAH1* and *RHOBTB3*) that had lost their DNA methylation in blastocysts continued to exhibit aberrant methylated in the E/A/Y/T tissues of cloned embryos (Fig. 5d and Supplementary Fig. 6b–e).

We then investigated whether abnormal DNA methylation exists during the in-vivo development of the cloned embryos. In line with the observations in E/A/Y/T tissues, the cloned monkey placentas also exhibited a globally higher level of methylation compared to the ICSI placentas (54.4% vs 48.3%, Fig. 5b, c middle). Significantly, we consistently observed a loss of DNA methylation in *DNMT1*, *THAP3*, *SIAH1* and *RHOBTB3* in SCNT placentas, indicating persistent demethylation of these genes in both pre- and post-implantation stages of SCNT embryos (Fig. 5e, f and Supplementary Fig. 7a–e). To ensure that the placenta tissue was free from contamination by maternal tissues, we isolated syncytial trophoblast cells (STB) from both ICSI and SCNT placenta based on their cell size. The WGBS analysis of these STB cells confirmed the consistent loss of DNA methylation in the imprinted genes from the blastocyst stage to the post-implantation stage (Supplementary Fig. 8). In addition to these four genes, *FGF12* and *JMJD1C* were also found to consistently lose DNA methylation from the blastocyst stage to the early post-implantation stage, as observed in the E/A/Y/T tissues, placenta tissues, and STB cells of cloned monkey embryos (Fig. 5e, Supplementary Figs. 6f, g and 7d, e and Supplementary Fig. 8f, g).

Both the hypoDMRs and hyperDMRs identified in blastocyst stage showed a much higher level of DNA methylation in the E/A/Y/T SCNT

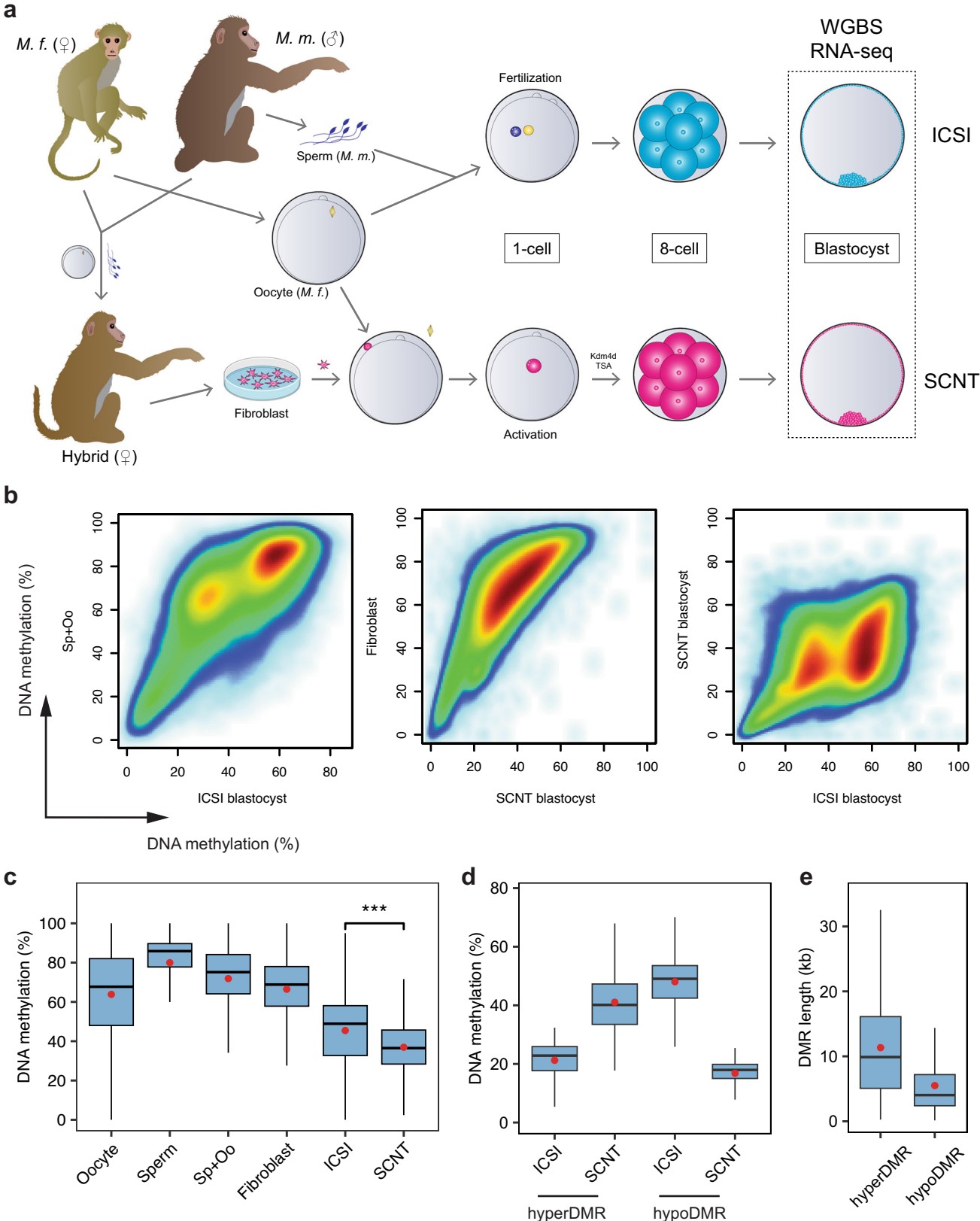

embryos (70.4%) compared to their ICSI counterparts (41.6%, Supplementary Fig. 6h). Although two out of three SCNT placentas displayed a similar DNA methylation level to that of the ICSI placentas in these blastocyst stage DMRs, we found that one cloned placenta showed ectopic higher methylation level than the others (Supplementary Fig. 6i), indicating heterogeneity within cloned monkey placenta.

### Defects of monkey SCNT placenta

Abnormal placenta development has been reported in numerous mammalians cloning studies[12,14,15,24]. To further explore if cloned placentas exhibited abnormal morphology, we conducted sonographic examinations and observed hyperplasia in the somatic cell cloned rhesus and cynomolgus monkey placentas (Supplementary Fig. 9a).

**Fig. 2 | Identification of DMRs between monkey ICSI and SCNT blastocysts.**
**a** Schematic diagram illustrating the production of hybrid ICSI and SCNT embryos. The abbreviation of *M.f.* indicates *Macaca fascicularis*, while *M.m.* is the short for *Macaca mulatta*. **b** Scatterplot comparing the DNA methylation levels between each sample. Pairwise comparison of the average DNA methylation levels in 10 kb windows. White represents regions with the lowest methylation, while red represents regions with the highest methylation. **c** The boxplot compares DNA methylation levels between monkey ICSI and SCNT blastocysts. Pairwise comparisons of average DNA methylation levels within 10 kb windows ($n = 292,059$). The upper and lower edges of the box represent the third and first quantiles, respectively, while the thick lines in the middle indicate the medians of each sample. The statistical analysis employed a two-sided Wilcoxon rank-sum test without adjustment,

resulting a $p$-value of <2.2e-16 (***). The red dots on the plot represent the means of each sample type. Two replicates were included for each sample ($n = 2$). ICSI, monkey ICSI blastocysts. SCNT, monkey SCNT blastocysts. **d** Boxplots displaying the DNA methylation levels of ICSI and SCNT blastocysts at hyperDMRs ($n = 9013$) and hypoDMRs ($n = 10,076$). The upper and lower edges of the box indicate the third and first quantiles, respectively. The thick lines in the boxes indicate the medians, and the red dots represents the means for each sample type. The number of DMRs is indicated in the figure. **e** The boxplot compares the lengths of hyperDMRs and hypoDMRs. The upper and lower edges of the box indicated the third and first quantiles, respectively, while the thick lines in the middle indicate the medians of hyper- and hypoDMRs. The red dots represent the means for each sample type.

Additionally, using Alizarin red (AR) staining, we detected aberrate deposition of calcium salts in some of the cloned placentas (Supplementary Table 3), while this was not observed in ICSI placentas (Supplementary Fig. 9b). To quantify the extent of calcification, we next counted the number of calcified clouds/dots in each placenta and found a higher average number of calcified clouds per mm² (CCN, Supplementary Fig. 9c, d, see Methods) in those highly calcified SCNT placentas. The average calcification index (CCI, percentage of area showing calcification, see Methods) for highly calcified SCNT placentas was $3.85 \pm 1.76\%$ (mean ± SD, $n = 3$ placentas, 7–37 slides/placenta), whereas the CCIs for placentas from ICSI and lowly calcified SCNT fetuses were $1.16 \pm 1.08\%$ and $0.59 \pm 0.26\%$, respectively (Supplementary Fig. 9e, f and Supplementary Table 3). In line with our sonographic findings, we also observed that the average thicknesses of SCNT placentas ($n = 8$ placentas, 3 for high calcified and 5 for low calcified) were more than 1.5-fold higher than those of ICSI placentas ($n = 4$ placentas) (Supplementary Fig. 9g, h and Supplementary Table 3).

### Full-term development of 2-cell fused monkey embryos
In mice, tetraploid complementation has been demonstrated to be an effective method for correcting placenta deficiencies and increasing the rate of full-term live births for SCNT embryos[24]. The generation of tetraploid embryos has also been reported in domestic animals such as rabbits, cows, and pigs[34–37]. However, it is yet to be determined whether tetraploid complementation can improve the efficiency of monkey SCNT.

Here, following a similar approach to the creation of tetraploid mice, we generated monkey tetraploid embryos through electrofusion (Supplementary Fig. 10a). At the blastocyst stage, were observed definitive ICMs in the electrofused monkey embryos (Supplementary Fig. 10a). Subsequently, we transferred a total of 49 2-cell fused monkey embryos into the uteruses of 16 surrogates, resulting in eight confirmed pregnancies with 16 implantation sites (Supplementary Table 4). Among the implanted embryos, eight developed into fetuses, while the remaining ones developed into gestational sacs only (Supplementary Fig. 10b and Supplementary Table 4). Finally, three out of eight fetuses were successfully born, with two monkeys currently surviving for over 2 years and one dying at around one year of age (Supplementary Fig. 10c and Supplementary Table 4). Karyotype analysis of the two surviving monkeys revealed normal chromosome numbers (Supplementary Fig. 10d).

To investigate why electrofused monkey embryos were able to develop into live fetuses, we performed live imaging using the electrofused embryos with H2B-EGFP overexpression. Our findings revealed two different types of mitosis models in these embryos (Supplementary Fig. 10e). In model I, the two nuclei initially come close to each other within the newly formed blastomere after the fusion of cleavage cells, and then divide into four new blastomeres, with each blastomere containing one nucleus. In this model, the two nuclei approach each other before undergoing the first mitosis. However, further investigation is needed to determine whether these two nuclei

undergo fusion to create a new, single nucleus or if they remain separate but closely juxtaposed prior to the M-phase. In Model II, despite the fusion of two cleavage cells and the formation of a new blastomere, the two nuclei do not come close to each other but independently undergo the first mitosis within the single fused blastomere, resulting in the formation of four blastomeres. These observations suggested the existence of a unique phenomenon in maintaining chromosomal ploidy stability in primate embryos, the specific mechanism of which is awaiting further exploration and verification.

### Rhesus monkey cloning with the trophoblast replacement method
Considering the limited success of tetraploid complementation in monkey embryos, we aimed to develop a TR method. In this method, the trophoblasts of SCNT embryos were replaced with those of ICSI embryos. First, the ICM of an SCNT blastocyst (SCNT-ICM) was obtained through immunosurgery. Then, the SCNT-ICM was injected into an ICSI blastocyst, from which the original ICM (ICSI-ICM) had been removed via micromanipulation immediately after the introduction of the SCNT-ICM (Fig. 6a–c, Supplementary Movie 1).

To ensure that there were no residual endogenous ICM remnants in the donor blastocysts, a total of 83 blastocoeles (ICSI blastocysts with ICMs removed) were transferred to the uterus without the injection of SCNT-ICMs (Supplementary Fig. 11a, Supplementary Table 6). In contrast to intact ICSI blastocysts, only 10 out of the 83 transferred embryos were successfully implanted. These implanted embryos developed into gestational sacs, but no live birth was observed by ultrasound imaging (Supplementary Fig. 11b–d, Supplementary Table 5). This result confirms the complete removal of endogenous ICMs from the blastocoele donor blastocysts.

We then combined the TR method with *Kdm4d* and TSA for rhesus monkey cloning. Out of 113 activated rhesus SCNT embryos, we transferred 11 reconstructed embryos to 7 surrogates (Table 1 and Supplementary Table 6). Through ultrasound examination, we observed that two of the surrogates became pregnant (Fig. 6d and Table 1), with one carrying twins. Unfortunately, the twin fetuses from the TR method were aborted on day 106 of gestation, and no fetal tissue was collected. However, a healthy male singleton was successfully born on day 157 and has survived well for over 2 years since then (Fig. 6e and Table 1). To confirm the mitochondria origin of this cloned monkey, we compared the SNPs in mitochondrial DNA (mtDNA) of the cloned monkey with those of the oocyte donor monkey (Fig. 6f and Supplementary Fig. 12). Furthermore, we performed short tandem repeat (STR) analysis on a total of 29 loci to verify the origin of the surviving monkey's genome. Remarkably, all loci demonstrated that the nuclear genome of the cloned monkey was identical to that of the fibroblast cell line used as the donor (rhMKEF180502-1) (Fig. 6g and Supplementary Table 7).

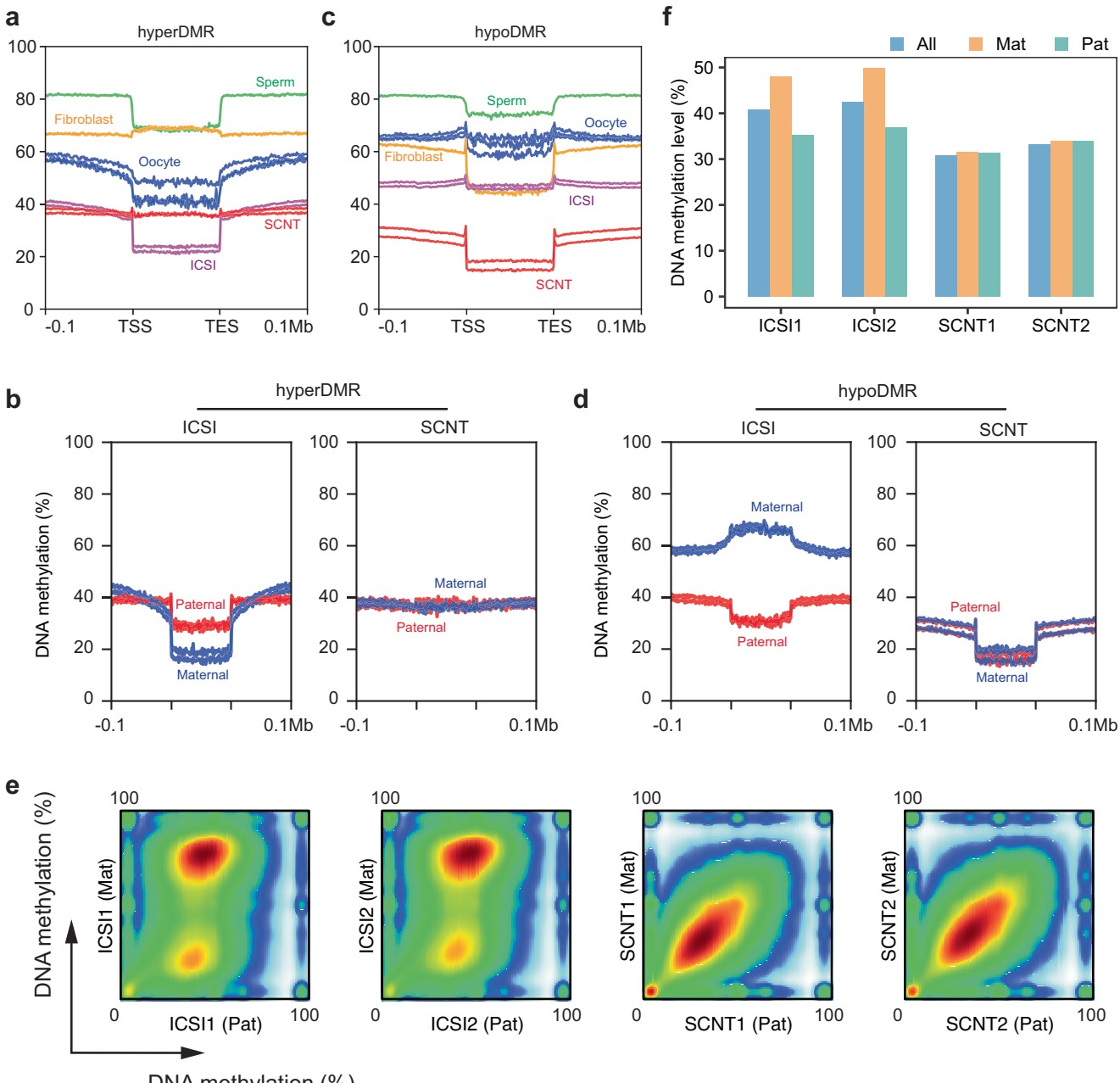

**Fig. 3 | Loss of maternally biased DNA methylation in monkey SCNT blastocysts. a** Comparison of average DNA methylation levels at hyperDMRs between monkey oocyte, fibroblast, ICSI blastocyst, SCNT blastocysts, and human sperm relative to their flanking regions. TSS refers to the transcription start sites, and the TES refers to transcription end sites. **b** Analysis of paternal and maternal allele-specific DNA methylation levels at hyperDMRs in ICSI and SCNT blastocysts, compared with their flanking regions. **c** Comparison of average DNA methylation levels at hypoDMRs between monkey oocyte, fibroblast, ICSI blastocyst, SCNT blastocysts, and human sperm, relative to their flanking regions. **d** Analysis of paternal and maternal allele-specific DNA methylation levels at hypoDMRs in ICSI and SCNT blastocysts, compared with their flanking regions. **e** Scatterplot comparing the DNA methylation levels between the paternal (Pat) and maternal (Mat) genomes of ICSI and SCNT blastocysts. **f** Bar plot comparing the DNA methylation levels of the whole genome (All), paternal (Pat) and maternal (Mat) genomes in ICSI and SCNT blastocysts.

## Examination of chimerism in the SCNT-TR monkey

One concern regarding the TR-derived SCNT monkey was the possibility of chimerism resulting from the ICSI-ICMs. To address this concern, we conducted an analysis of SNPs on the X chromosome, given that a male donor cell line (rhMKEF180502-1) was used as the nuclear donor (Fig. 7a).

For the rhesus clone, the employed Sanger sequencing of the two specific X chromosome SNPs to demonstrate the complete SCNT-ICM origin of the fetus tissues and the ICSI embryo origin of the TR-derived placenta (Fig. 7b). To further confirm this, we subjected ear tissue, skin, and peripheral blood samples, which were derived from the ectoderm and mesoderm layers, to site-specific deep sequencing of their genomes. The results confirmed that the ICSI-ICM did not contribute to fetal development (Fig. 7c).

Moreover, our findings revealed that the rhesus TR-placenta inherited X chromosomes from both the oocyte donor and sperm donor for ICSI, while only one copy of the X chromosome was detected in the ectoderm and mesoderm layers of the rhesus SCNT-TR monkey (Fig. 7d). This implies that a female placenta supported the full-term development of a male SCNT fetus (Fig. 7e, f). It is

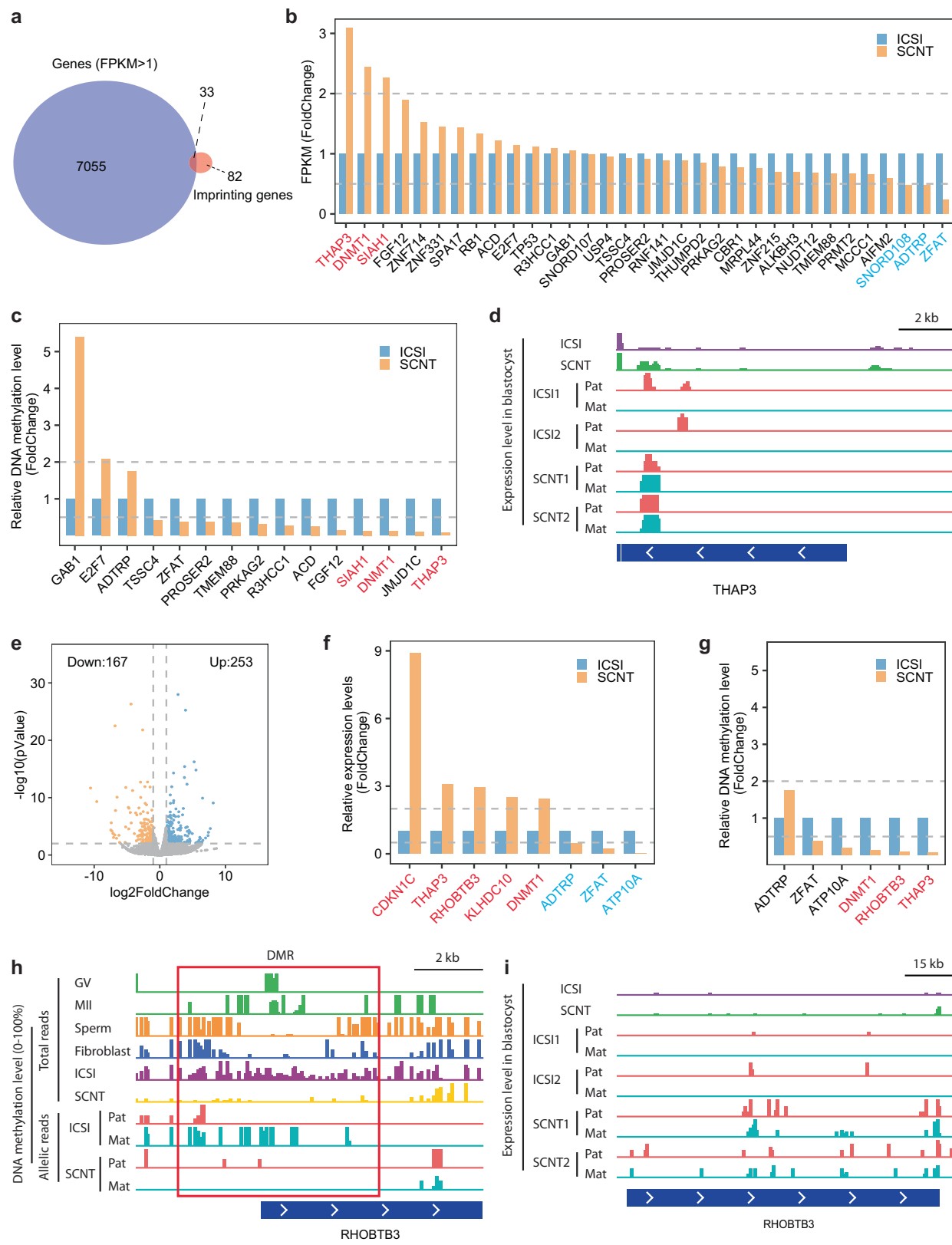

worth noting that the placenta of this cloned rhesus monkey consisted of cells from ICSI and SCNT embryos (Fig. 6g and Supplementary Table 7), indicating that the injected SCNT-ICM contributed partly to the placenta of the reconstructed embryo.

## Morphology of the SCNT-TR placenta

Through a comparison of the WGBS data, we observed a similar DNA methylation level between SCNT-TR and ICSI placentas (Fig. 5b, c right). Additionally, we discovered that genes such as *DNMT1*, *THAP3*,

**Fig. 4 | Identification of aberrantly expressed imprinted genes in monkey SCNT blastocysts. a** Venn diagram illustrating the filtration process of imprinted genes using FPKM. We obtained a total of 33 candidate imprinting genes as indicated in the figure. **b** Bar graph displaying the relative gene expression levels of 33 imprinted genes reliably detected in both ICSI and SCNT blastocysts (FPKM > 1). The gene expression level of ICSI blastocysts was set as 1. **c** Bar graph showing the relative DNA methylation levels of 15 (out of 33 in Fig. 4b) genes that correlated with the DMRs. Genes in red were identified as candidate ectopically expressed imprinted genes due to the loss of DNA methylation. The DNA methylation level of ICSI blastocysts was set as 1. **d** Genome browser view illustrating the expression levels of *THAP3* in ICSI blastocyst, SCNT blastocyst, and their parental genomes. Mat maternal allele. Pat paternal allele. Scale bar, 2 kb. **e** Volcano plot presenting the differentially expressed genes (DEGs) between SCNT and ICSI blastocysts (SCNT vs. ICSI). Up in the figure refers to upregulated genes, while down denotes

those downregulated genes. **f** Bar graph showing the relative gene expression of 8 imprinted genes identified as DEGs between ICSI and SCNT blastocysts (*p*-value < 0.01, fold change >2 and baseMean >10). The expression level of ICSI blastocysts was set as 1. Genes upregulated and downregulated in SCNT embryos are colored red and blue, respectively (fold change >2). **g** Bar graph displaying the relative DNA methylation levels of 6 (out of 8 in Fig. 4f) genes that correlated with DMRs. Genes in red were identified as candidate ectopically expressed imprinted genes due to loss of DNA methylation. The DNA methylation level of ICSI blastocysts was set as 1. **h** Genome browser view illustrating the DNA methylation levels of *RHOBTB3* and its flanking regions. The red box represents the identified DMR of *RHOBTB3*. Mat, maternal allele. Pat, paternal allele. DMR differentially methylated regions. Scale bar, 2 kb. **i** Genome browser view illustrating the expression levels of *RHOBTB3*. Mat maternal allele. Pat paternal allele. Scale bar, 15 kb.

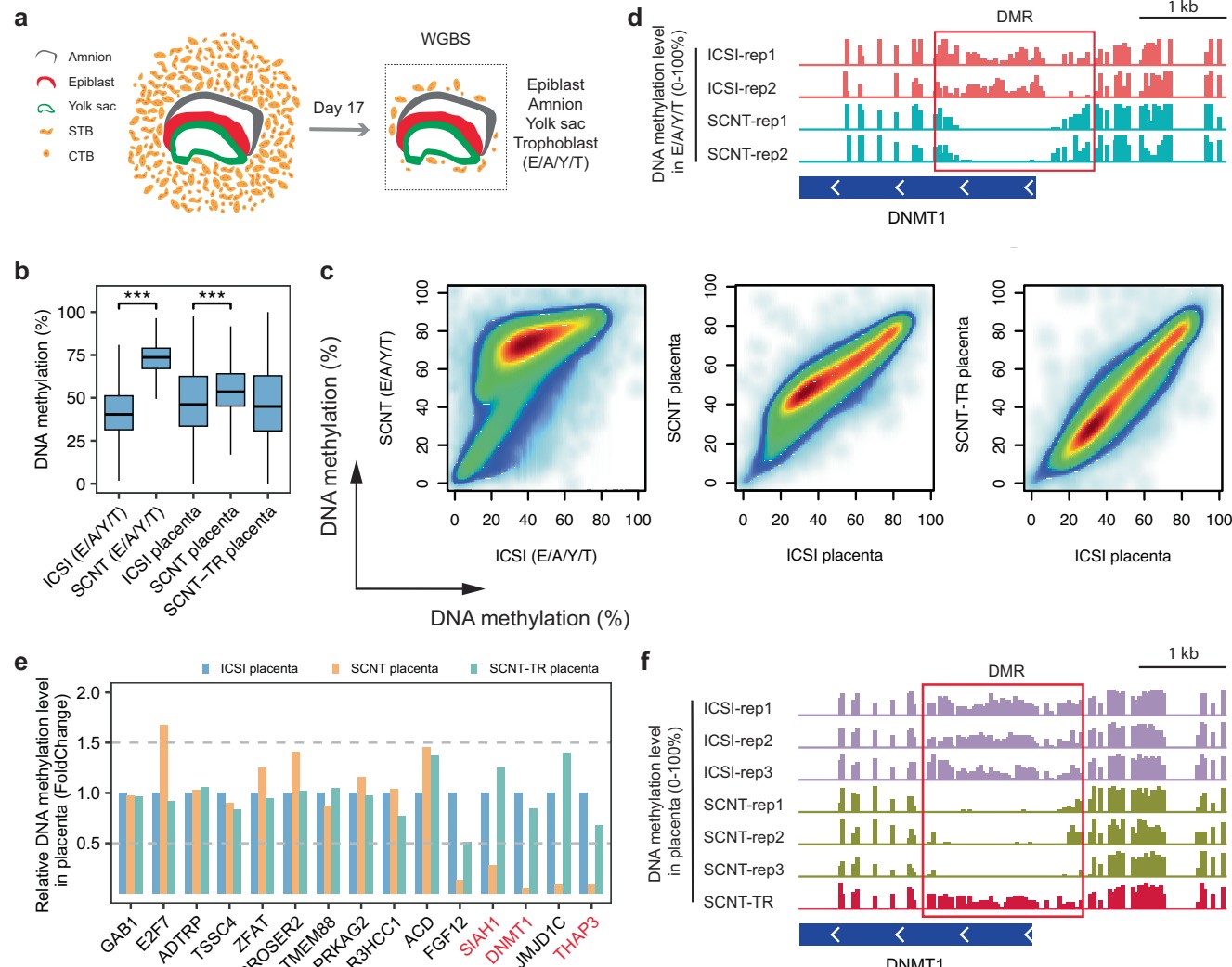

**Fig. 5 | Persistent loss of imprinting in post-implanted monkey SCNT embryos. a** Diagram illustrating the isolation of E/A/Y/T tissues from in vitro cultured post-implantation embryos. **b** Box plot displaying the global DNA methylation levels of ICSI, SCNT E/A/Y/T tissues, as well as ICSI, SCNT and SCNT-TR placentas. The statistical analysis was performed using two-sided Wilcoxon rank-sum test without adjustment. *p*-value < 2.2e-16 (***). The upper and lower lines of the blue boxes indicate the third and first quantiles, while the thick lines in the middle means the median of each sample. Whole genomes of each sample were divided by a 10-kb step, and a total of 292,059 windows for each sample were obtained (*n* = 292,059).

**c** Scatter plot comparing the DNA methylation levels of post-implanted ICSI and SCNT embryos (left), ICSI and SCNT placentas (middle), and ICSI and SCNT-TR placentas (right). **d** Genome browser view showing the continuous loss of DNA methylation in *DNMT1* in cloned embryos at the post-implantation stage. **e** Bar plot showing the DNA methylation levels of the DMRs of imprinting genes (Fig. 4c) in ICSI, SCNT and SCNT-TR placentas. **f** Genome browser view revealing the persistent loss of DNA methylation in *DNMT1* in cloned placenta and the restoration of DNA methylation in SCNT-TR placenta.

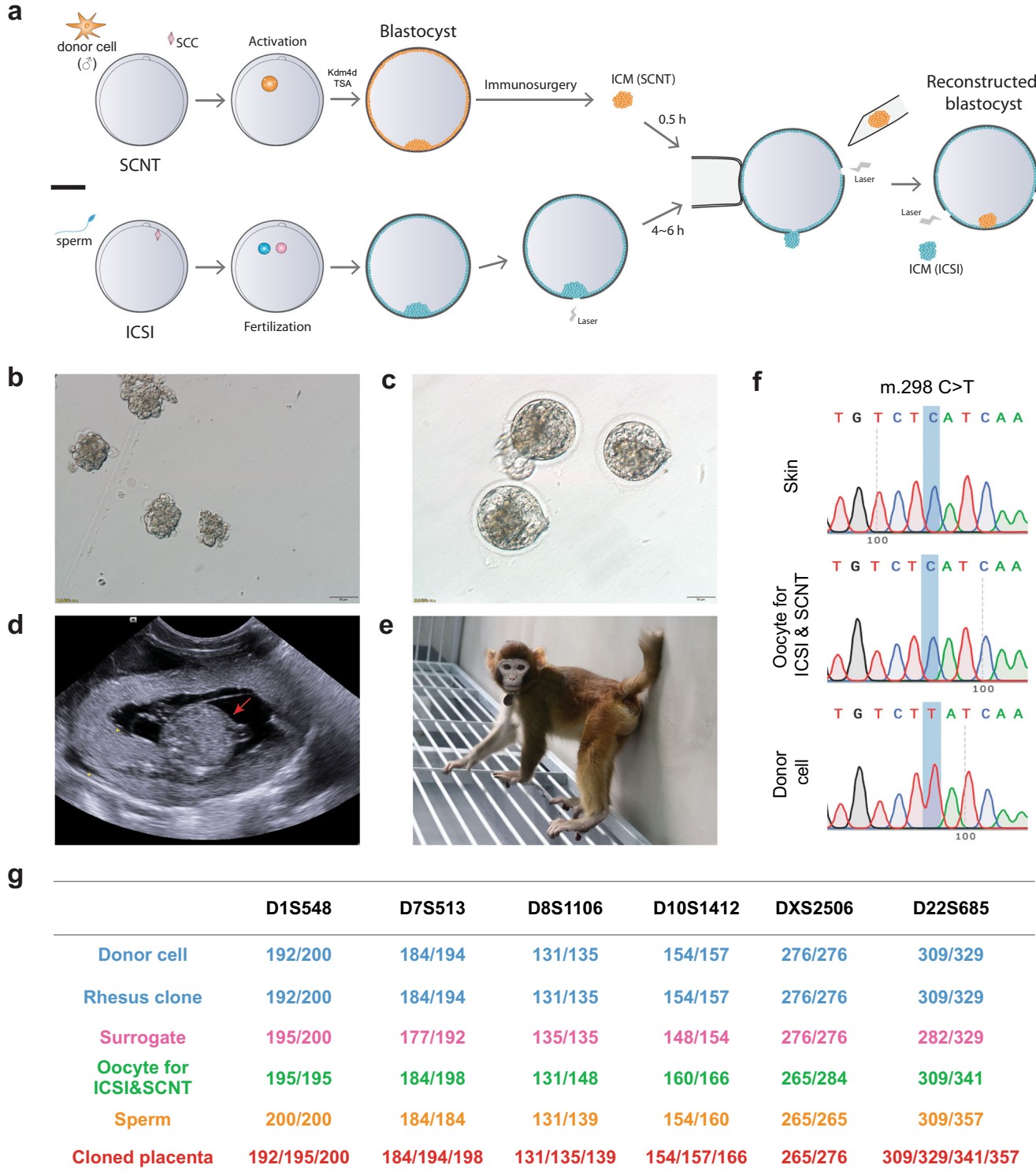

**Fig. 6 | The live birth of a somatic cell-cloned rhesus monkey through trophoblast replacement. a** Diagram illustrating the trophoblast replacement procedure between monkey ICSI and SCNT blastocysts. SCC in the figure is the short for spindle-chromosome complex. **b** Close-up image of the inner cell mass of the monkey SCNT blastocysts. This experiment was independently repeated for 5 times in this study. Scale bar, 50 μm. **c** Image of the reconstructed blastocysts after injecting SCNT-ICMs into the ICM-free ICSI blastocoeles. This experiment was independently repeated for 5 times in this study. Scale bar, 50 μm. **d** Ultrasound examination of the SCNT-TR rhesus fetus on gestation day 60. **e** Photograph of the somatic cell-cloned rhesus monkey produced through trophoblast replacement, taken at 17 months. **f** An example of a single nucleotide polymorphism (SNP) allele indicating the mitochondrial DNA (mtDNA) origin of the cloned rhesus monkey. **g** Representation of short tandem repeat (STR) loci (D1S548, D7S513, D8S1106, D10S1412, DXS2506, D22S685) in the cloned rhesus monkey.

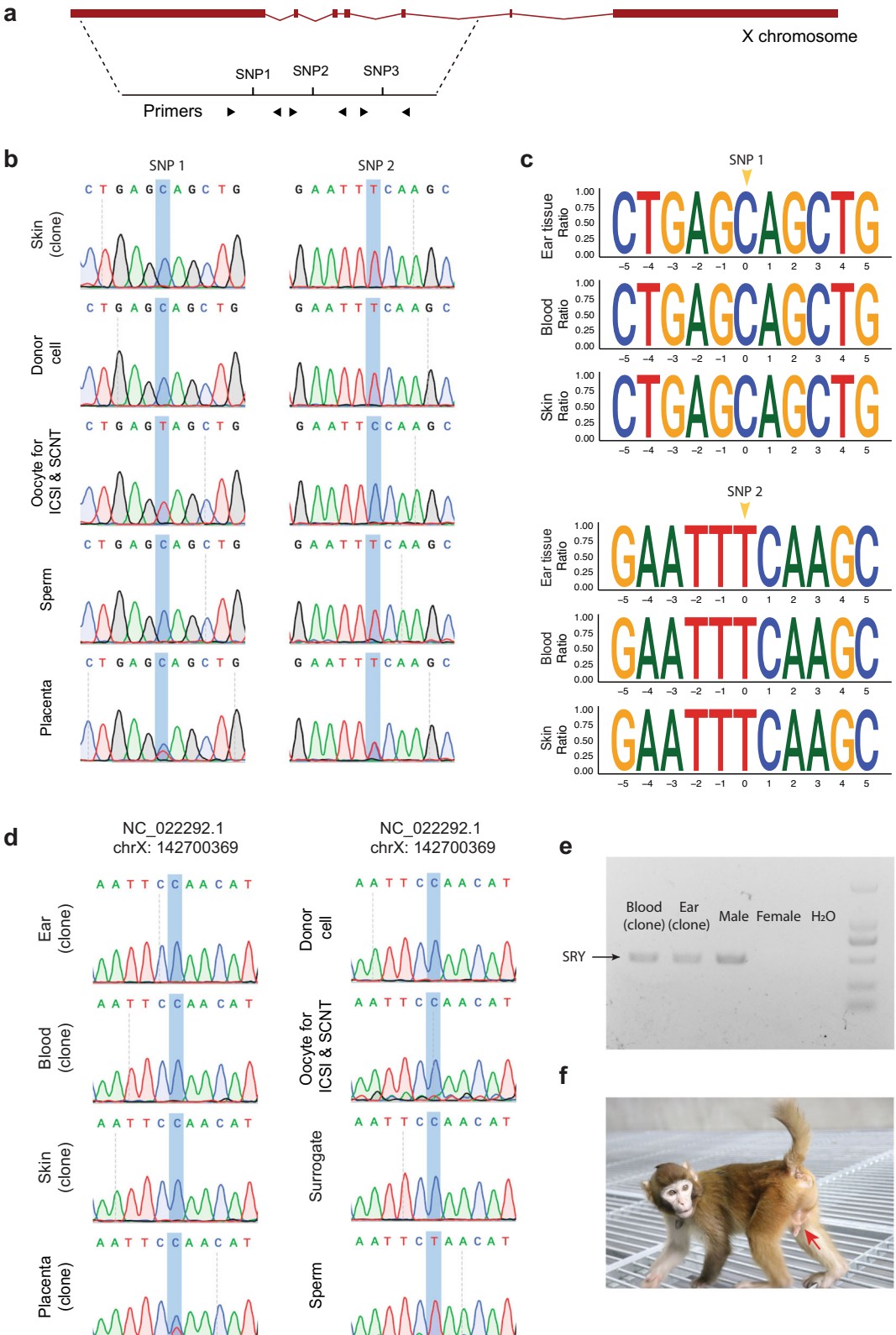

**Fig. 7 | Chimerism examination of the cloned rhesus monkey. a** Diagram out-lining the strategy for examining chimerism. **b** Genotype analysis of two X chromosome SNPs (SNP1 and SNP2 in Fig. 7a) in the cloned monkey (skin), donor cell line, oocyte donors, sperm donor, and the rhesus TR placenta. **c** Deep sequencing of two X chromosome SNPs at specific genome sites using peripheral blood, skin, and ear tissues of the cloned rhesus monkey. More than 10,000 reads per sample per locus were obtained. The SNP sites are marked with yellow arrowheads. **d** Identification of a SNP (SPN3 in Fig. 7a) indicating the inherited X chromosome from the sperm donor in the TR placenta. **e** PCR analysis of SRY using peripheral blood and ear tissue of the cloned rhesus monkey. This experiment was independently repeated for twice in this experiment. **f** Image of the cloned rhesus monkey's genital, highlighted by a red arrow.

*SIAH1*, *RHOBTB3*, *FGF12*, and *JMJD1C*, which experienced persistent loss of DNA methylation, were rescued in the SCNT-TR placenta (Fig. 5e, f and Supplementary Fig. 7a–e). The hypo- and hyperDMRs were also found to be corrected in the SCNT-TR placenta (Supplementary Fig. 6i).

Subsequently, we compared the CCI and CCN of the SCNT-TR placenta with those of ICSI and SCNT placentas. Significantly lower CCI and CCN were observed in SCNT-TR placentas compared to that of highly calcified SCNT placentas (Supplementary Fig. 9i, c–f, Supplementary Table 3). Since SCNT-TR placentas exhibited similar calcification levels as the ICSI placentas, we concluded that the prevention of premature calcification could enhance the post-implantation development of the cloned fetus. Nevertheless, the thickness (17.39 mm) of the SCNT-TR placenta remained similar to that of the SCNT placentas for reasons yet to be determined (Supplementary Fig. 9g, h, Supplementary Table 3).

Taken together, our findings demonstrate that TR can restore the abnormal loss of DNA methylation in the placenta and prevent highly calcified phenotypes in monkey SCNT placentas. Although further examination is needed to assess the potential of this method in improving the cloning efficiency, we have successfully achieved the birth of a healthy live cloned rhesus monkey through SCNT.

## Discussion

The fact that nearly half of the cloned monkey fetuses were lost before day 60 of the gestation period suggests that the SCNT embryos were defective during implantation. In a recent study, scientists discovered aberrant gene expression of imprinted genes, including *DNMT1*, *PCLAF*, *SIAH1*, and *RHOBTB3*, in rhesus monkey SCNT placentas[38]. Here in this study, we dissected the somatic cell reprogramming mechanism more detailly by integrating the WGBS and RNA-seq datasets of both the pre- and post-implanted monkey embryos.

The WGBS examination of hybrid (*Macaca fascicularis* × *Macaca mulatta*) ICSI embryos and hybrid (*Macaca fascicularis* × *Macaca mulatta*) somatic cell-derived SCNT embryos revealed the presence of inherited DMRs from fibroblasts to the SCNT blastocyst. We also identified the inherited loss of maternal imprinting for several imprinted genes in both pre- and post-implanted monkey SCNT embryos by integrating WGBS and RNA-seq data. Out of the 4 identified genes in this study, three (*DNMT1*, *THAP3* and *SIAH1*) were known to be imprinted in the human placenta[39,40], while *RHOBTB3* was identified as paternally-biased expression in human placenta[41,42]. In a recent study, the upregulation of *SIAH1* was observed in spontaneous abortion patients and resulted in impaired invasion and proliferation of the trophoblast of human embryos[43]. Additionally, two other genes, *FGF12* and *JMJD1C*, showed persistent loss of DNA methylation in cloned blastocyst and post-implanted embryos. However, the role of these two genes during somatic cell reprogramming requires further investigation as their gene expression levels showed no alteration in the somatic cell cloned embryos. Interestingly, the loss of DNA methylation modifications in all of these genes was detected in somatic cell but not in post-implanted ICSI embryos (day 17 post fertilization). The complete reprogramming of these DMRs and the precise function of maintaining these DMRs from oocyte to early post-implant stages and placentas need to be thoroughly explored in future studies. On another note, although *CDKN1C* and *KLHDC10* were not associated with DMRs, they were found to be upregulated DEGs in this study. Increased *CDKN1C* mRNA levels have been observed in cases of the intrauterine growth restriction in both mice and human[32,44]. *KLHDC10* has been reported as a maternally-biased gene in its expression in the human placenta[30,41,45]. Taken together, the presence of these ectopically imprinted genes in the SCNT blastocysts suggested the inheritance of epigenetic abnormalities from the somatic cells during SCNT.

In the SCNT trophoblast DNA methylation data, we noticed that replicate one showed complete loss of DNA methylation at the DMR of

*THAP3*, *DNMT1*, *RHOBTB3* and *SIAH1*, while replicate two showed partial preservation DNA methylation at these loci (Supplementary Fig. 5g–j). This suggestion that DNA methylation at these loci in the donor cells for replicate two SCNT sample might have escaped the DNA methylation loss at these loci. Therefore, somatic cells that exhibit better maintenance of DNA methylation at the imprinting DMRs are potentially a better source of donor cells for more efficient monkey cloning, which represent a promising direction for future exploration.

Misexpression of imprinted genes disrupts regulatory networks and leads to abnormalities in embryonic development. The imperfect reprogramming process in SCNT may cause epigenetic changes in imprinted genes, contributing to developmental arrest at early implantation stages. Additionally, imprinted genes are involved in placental development, which is often impaired in SCNT embryos. During NHP cloning, we observed abnormal morphological development of SCNT placentas, specifically severe calcification. The underlying mechanism for this aberration and its impact on embryo development failure are yet to be determined. We believe that further molecular investigation is needed to understand the obstacles to post-implantation development of monkey SCNT embryos.

While tetraploid complementation improved the efficiency of mouse SCNT[24], a similar approach may not be applicable to NHPs, as our study showed that the electrofused 2-cell stage embryos developed to full-term after implantation. Here, we developed a TR method for NHP SCNT by modifying the previously reported methods for transferring exogenous ICMs into intact sheep, cow, and monkey blastocysts[22,23,46]. In simple terms, we used laser assistance to remove the endogenous ICMs rather than cutting them and the surrounding cells with a microblade. The reconstructed blastocysts still retained the zona pellucida to a large extent. Moreover, our method allowed for the expansion of the blastocoele while the endogenous ICM was hatched out of the zona pellucida, ensuring the complete removal of the endogenous ICM. By combining this TR method with a previous SCNT protocol[8,9], we successfully cloned a rhesus monkey that has survived for over 2 years. As a result of embryo loss during the TR procedure, we achieved a live fetus out of 113 (1/113) activated SCNT embryos. However, when considering only the transferred embryos, we obtained 1 live fetus out of 11 (1/11) transferred embryos. It is important to note that the calculation of the live birth efficiency is conducted based on embryos activated. This is because TR must be performed at the blastocyst stage, and none of the non-TR embryos were transferred at the blastocyst stage. Therefore, the consideration is focused on embryos activated at this specific situation rather than later-stage embryos.

Lastly, the TR technique involves replacing the dysfunctional or defective trophectoderm cells with healthy ones, which ultimately rescuing the development of fertilized embryos. This strategy holds great promise for improving the success rates of by addressing issues specifically related to the trophectoderm, which plays a crucial role in early embryonic development and implantation.

By implementing this technique in conventional IVF, embryos with abnormalities or deficiencies in the trophectoderm can be rescued. For instance, embryos exhibiting compromised implantation potential, or abnormal gene expression patterns in the trophectoderm can benefit from this innovative approach. It offers a means to circumvent limitations associated with embryonic development, preventing the implantation of embryos with compromised trophectoderm function and increasing the chances of successful pregnancy outcomes.

Furthermore, the TR technique may also have broader implications in the field of assisted reproductive technologies. This advancement opens the door for further research and exploration into optimizing the trophoblast cell therapy approach, potentially extending its application to other types of infertility, such as recurrent

implantation failure and certain genetic disorders affecting the trophectoderm.

## Methods

### Animal ethics statement

The monkeys utilized in this research were sourced from the NHP Facility of the Center for Excellence in Brain Science and Intelligence Technology in Shanghai, China. All animal procedures adhered to the guidelines set by the Animal Use and Care Committees at the Shanghai Institute of Biological Science, Chinese Academy of Sciences (CAS), and the Institute of Neuroscience, CAS Center for Excellence in Brain Science and Intelligence Technology. The approved application by the committees is titled "The research of constructing macaque animal model by SCNT" (ION-2018002R01).

### Superovulation and oocyte collection

The procedures for superovulation and oocyte collection were conducted according to our previously reported method[8,9]. In summary, the oocyte donor monkeys were initially primed with human follitropin and chorionic gonadotrophin. The oocytes in the follicular fluid were then aspirated into pre-warmed HEPES-buffered Tyrode's lactate medium (TH3) supplemented with heparin. Subsequently, the oocytes were cultured in pre-equilibrated 1066 medium to achieve further in vitro maturation[47,48].

### Fibroblast isolation and culture

The preparation of primary fibroblasts as donor cells was performed as described previously[8,9]. In brief, skin tissue of a 62-day-old rhesus monkey fetus was cut into small pieces using sterile scissors. The tissue was then digested in DMEM containing DNase (1 mg/ml), collagenase IV (0.5 mg/ml), 100 IU/ml penicillin and streptomycin, 10% fetal bovine serum (FBS), 1% non-essential amino acids, and 1% glutamax. The digestion process took place at 37 °C in 5% $CO_2$ for 4 h. The dissociated fibroblasts were allowed to proliferate in culture dish until they reached confluency, which typically took for 10–20 h. The cultured fibroblasts were then cryopreserved in a medium consisting 10% dimethylsulfoxide and 90% FBS for future use. Thawed fibroblasts were further cultured for several days prior to performing the nuclear transfer.

As for the culture of fibroblasts derived from electrofused embryos, the skin tissues were obtained and washed in PBS containing 1% penicillin-streptomycin, repeating this process three times. The tissues were subsequently cut into small pieces and implanted onto culture dishes. Notably, before implantation, the 6-cm culture dishes were treated with 1 ml of fibroblast culture medium before the skin tissues were implanted. After implantation, the culture dishes were incubated at 37 °C, 5% $CO_2$ and 1 ml of culture medium was added every other day until the fibroblasts were observed (about 1 week).

### Monkey SCNT and ICSI

The procedures for monkey SCNT and ICSI were carried out according to previously reported[8,9]. Briefly, metaphase II (MII)-arrested oocytes were selected for both SCNT and ICSI. For SCNT, 10–20 MII oocytes were placed into TH3 medium containing 5 mg/ml cytochalasin B in a glass-bottomed dish. The spindle-chromosome complexes of the oocytes were removed, and HVJ-E (Cosmo Bio Co.Ltd, 808920) virus-treated donor fibroblasts were injected into the perivitelline space of the enucleated oocytes. After 1–2 h of fibroblast-oocyte fusion, the resulting SCNT embryos were activated using ionomycin and 6-dimethylaminopurine. At 5–6 h after activation, human *Kdm4d* was injected into the SCNT embryos. The embryos were then treated with TSA (Sigma, T1952) for 10 h starting from their artificial activation. As for ICSI, a single sperm was injected into the cytoplasm of MII oocytes, and fertilization was confirmed by the appearance of two pronuclei approximately 6 h after injection. In order to synchronize the development of both types of embryos, TR was performed on the same day using the same batch of collected oocytes.

### Immunostaining of the monkey SCNT embryos

The evaluate the stability of chromosome behavior during the first mitosis of SCNT embryos, immunostaining was performed to examine both chromosomes and tubulin organization. In brief, 1-cell stage embryos with observable nuclear envelop break down were collected and fixed using 4% paraformaldehyde (PFA) for 30 min at room temperature. Following fixation, the embryos were washed three times with PBS containing 0.1% BSA. Subsequently, the embryos were permeabilized using 0.5% Triton X-100 and blocked using 5% BSA for 3 h.

Next, the embryos were incubated overnight at 4 °C with the primary antibodies, followed by three times wash with PBS (each wash lasting 10 min). Subsequently, the embryos were incubated with the secondary antibodies for 2 h in 37 °C. After washing, the stained embryos were placed in a glass-bottomed dish (WPI, FD3510-100) containing PBS drops covered with paraffin oil. The embryos were finally imaged using FV3000 (Olympus), and Z-stack confocal imaging was performed with a 1-μm interval.

The final concentrations of the primary antibodies used in this study were as follows: anti-$\partial$-tubulin (1:200, abcam, ab80779), anti-pH3 (1:100, abcam, ab5176). The final concentrations of the second antibodies and DAPI used in this research were as follows: Alexa Fluor® 488-conjugated AffiniPure Donkey Anti-Mouse IgG (H + L) (1:5000, JackSon ImmunoResearch, 715-545-150), Alexa Fluor Cy3-conjugated AffiniPure Donkey Anti-Rabbit IgG (1:5000, JackSon ImmunoResearch, 711-165-152) and DAPI (1:5000).

### Electrofusion of two-cell stage monkey embryos

The protocol for monkey tetraploid embryo creation was performed as described previously[49]. The 2-cell stage monkey ICSI embryos were washed three times in pre-warmed TH3 and then washed another three times with 0.3 mol/l D-Mannitol (Sigma, M4125). Subsequently, the embryos were placed in the D-Mannitol solution, and cell fusion was induced using electric pulses (30 V, 40 μs, two pulses) delivered by the Eppendorf Multiporator 4308 Electroporation System. The embryos were then incubated at 5% $CO_2$, 37 °C for 2 h to allow cell fusion to occur.

To examine nuclear fusion, mRNA of H2B-EGFP (Addgene, 53744) was injected into 1-cell stage embryos and allowed to express until the embryos reached the 2-cell stage. After electrofusion, the embryos were observed using a Confocal imaging system (FV10i, Olympus) during the first mitosis.

### Karyotype analysis of monkeys derived from 2-cell stage electrofused embryos

The karyotype analysis was conducted following a previously described method with minor modifications[9]. Fibroblasts were initially treated with Colchicine (100 ng/ml, APExBio, A3324) for 12 h. After washing, the cells were dissociated using 0.25% Trypsin-EDTA and exposed to a pre-warmed hypotonic solution for 30 min at 37 °C. Subsequently, 2–3 ml pre-chilled fixation solution was gently added to the cell and hypotonic solution mixture, and the cells were pre-fixed at room temperature for 10 min. The cells underwent centrifugation and were resuspended with fresh pre-chilled fixation solution three times for proper fixation. Finally, the cells were dropped onto slides and stained with Giemsa (Sigma, GS500). The resulting slides were then observed and imaged using VS120 microscope (Olympus).

### Trophoblast replacement

First, a small slit was created by laser ablation on the polar side of an ICSI blastocyst to penetrate the zona pellucida. After about 6 h of culturing, the ICSI-ICM selectively hatched out of the zona pellucida through the slit, while the blastocoele remained expanded. About 5 h

after creating the slit on ICSI blastocysts, the SCNT-ICM was obtained through immunosurgery. The isolated SCNT-ICM was then immediately injected into the ICSI blastocoele after another slit was made. Finally, the hatched endogenous ICSI-ICM was removed through laser irradiation. Typically, the ratio of SCNT-ICM to ICSI trophoblast blastocoele was 1/1, although in some cases two to three SCNT-ICMs were injected into one blastocoele. It is worth noting that the cloned monkey obtained in this research were derived from a 1/1 injection (Supplementary Table 6).

The immunosurgery was performed as described previously with minor modification[46]. Briefly, the zona pellucida of the SCNT blastocyst was dissolved using pre-warmed acid Tyrode's solution (Sigma, T1788), and the blastocysts were then washed three times in H9 medium. Subsequently, the blastocysts were cultured in rabbit anti-monkey IgG whole serum (Bioss, bs-0335Rs) for 30 min, and after three additional washes in H9 medium, they were transferred into Guinea pig complement serum (Fitzgerald, 32R-CP004) for another 30 min. The intact SCNT-ICMs were isolated by gentle pipetting with a beveled pipette (around 50 μm in diameter) in TH3 medium under micromanipulation. The SCNT-ICMs were immediately injected into the ICSI blastocoeles following isolation.

### Monkey embryo culture and transfer
Monkey embryos were cultured at 37 °C in 5% $CO_2$ in serum-free H9 medium until they reached the 8-cell stage (day 3 after fertilization). They were then transferred to H9 medium supplemented with 5% embryonic stem cell grade FBS. The medium was changed every 2 days until the embryos reached the blastocyst stage.

Within 2 h of reconstruction, the reconstructed embryos were transferred into the oviducts of surrogates in which fresh stigmas were observed on the ovaries.

### STR analysis
Ear tissue from the cloned rhesus monkey, along with donor fibroblasts and peripheral blood samples from the oocyte donor, sperm donor, and surrogate mother were collected for STR analysis. For PCR amplification, locus-specific primers were used, with each primer tagged with a fluorescent dye (FAM/HEX/TMR). The amplified STR products, labeled with FAM-, HEX-, or TMR, were then diluted and mixed with an internal size standard ROX500 and deionized formamide. Capillary electrophoresis was performed on an ABI PRISM 3730 genetic analyzer to obtain the raw data. The obtained raw data was analyzed using the Gene Marker 2.2.0 program.

### mtDNA examination
Total genomic DNA was extracted from ear tissue or blood samples of the cloned monkey, surrogate mother, sperm donor, and oocyte donor, as well as DNA from donor cells, using the Genomic DNA Extraction Kit following the manufacturer's instructions (Tiangen, DP304-03). To identify SNPs among the samples, the entire mtDNA sequence was amplified using 13 pairs of primers. The DNA amplification involved 35 cycles of denaturation at 95 °C for 30 s, annealing at 57 °C for 30 s, and extension at 72 °C for 1–1.5 min. This was followed by a 5-min extension at 72 °C. The PCR products were then subjected to sequencing, and the obtained sequences were used for the SNP analysis.

### Chimerism examination of the cloned rhesus monkey
Chimerism examination was conducted following the previously described method[50]. The genomes of the cloned monkey, donor cell, oocyte donors for ICSI and SCNT, placenta, sperm donor, and surrogate were amplified and sequenced for SNP detection. After SNP detection, site-specific deep sequencing primers (with barcoding primers added) were used to amplify the genomes. The PCR products were then purified using the Monarch PCR & DNA Cleanup Kit (New

England Biolabs), and data analysis was carried out using the Hi-TOM platform. The filtration threshold was set at 1% with a minimum of 10,000 reads per sample. The primers used for X chromosome SNP detection and site-specific deep sequencing were listed in Supplementary Table 8.

### In vitro transcription of *Kdm4d*
For in vitro transcription of *Kdm4d*, the DNA template was amplified using a primer that include a T7 promoter. The resulting T7-*Kdm4d* PCR product was purified and utilized as the template for in vitro transcription using the mMESSAGE mMACHINE T7 ULTRA kit (Life Technologies, AM1345). The RNA products were then purified using the MEGA clear kit (Life Technologies, AM1908) and eluted in RNase-free water.

### In vitro transcription of H2B-EGFP
For the in vitro transcription of H2B-EGFP, the coding sequence (CDS) was first cloned from the plasmid (Addgene, 53744) using PCR. The transcription process was carried out using the mMESSAGE mMACHINE™ SP6 Transcription Kit (Life Technologies, AM1340). Subsequently, the RNA products were purified using MEGA clear kit and eluted in RNase-free water.

### Alizarin red staining
The whole placentas were fixed in 4% paraformaldehyde for a duration of 1 month before being sectioned. Due to their large size, the placentas were diced and the tissues were embedded in pre-melted paraffin. After three rounds of paraffin embedding, the tissues were placed into paraffin block molds and allowed to cool. After cooling, the tissues were sectioned into 5-μm thick sections and floated on clean water at room temperature and then in a 45 °C water bath to allow them to flatten. The sections were then gently transferred onto glass slides to dry for 1–2 h. To melt the wax, the slides were placed in a 60 °C drying oven for 30 min. Once cooled, the slides underwent dewaxing in xylene and a decreasing concentration gradient of ethanol (100%, 90%, 80%, and 70%). After rinsing with water, thee slides were ready for AR staining.

For the AR staining procedure, the slides were treated according to the manufacturer's instructions (Servicebio, G1038). In brief, the slides were incubated in AR solution for 5 min, followed by three washes in PBS. After drying for 4 h at 60 °C, the slides were incubated in xylene for clearing. Finally, the tissues were mounted with synthetic resin and covered with coverslips. The slides were left to dry for 3 days at room temperature before being imaged with VS120 (Olympus) under brightfield.

### Morphometry of placenta calcification
The AR-stained slides were utilized for the observation of calcification. The calcified area and total area of each placenta were determined using ImageJ. Specifically, each slide was converted to 8-bit color mode, then the "Huang" and "Yen" methods were selected for obtaining the total area and the calcified area, respectively. Notably, the particles smaller than 0.0003 $mm^2$ were filtered out, and all of the slides were analyzed with default parameters. The particle areas of the placenta sections and the calcified regions of each slide were obtained with the default thresholds of the "Huang" and "Yen" methods.

The CCI was calculated as follows:

CCI = (total calcified area of each placenta/total area of each placenta)*100%

The average CCN was calculated as follows:

CCN = (number of total calcified dots in each placenta/total area of each placenta)*100%

The average thickness of each placenta was measured using ImageJ. Briefly, the thickness of the placenta sections on each slide was

measured and the average thickness of each placenta was calculated as follows:

Thickness=the sum of the thickness of each placenta/number of slides per placenta

## WGS data processing

The parental genomes were extracted from the blood samples of the parents of hybrid embryos (Supplementary Table 1). The sequenced raw reads were trimmed using fastp with default parameters to remove low-quality reads and adapters in paired-end reads[51]. The remaining clean reads were aligned to the *Macaca fascicularis* genome (macFas5) using the "mem" mode of BWA[52] in default parameters. The read duplicates were marked with the MarkDuplicates mode of GATK[53] with default parameters. Haplotyper and GVCFtyper were performed using sentieon[54] and GTAK. SNP calling and filtering were performed using the "SelectVariants" of GATK (version 4.2.6.1). Then, SNPs were filtered using the "VariantFiltration" module of GATK with parameters: $QD < 2.0 || MQ < 40.0 || FS > 60.0 || SOR > 3.0 || MQRankSum < -12.5 || ReadPosRankSum < -8.0$. The BED files were generated by selecting only the homologous SNP positions between maternal genomes and the paternal genome. Next, the parental genomes were masked by the BED files containing the known homologous SNP positions using BEDtools[55].

## Collection of monkey blastocysts and trophoblasts

The zona pellucida of blastocysts was dissolved after treatment with acid Tyrode's solution to remove the surrounding somatic cells. Then, the blastocysts were thoroughly washed in prewarmed PBS containing 0.1% BSA. The resulted blastocysts were then transferred into freshly prepared PBS and immediately put into liquid nitrogen for fast frozen.

The trophoblasts were collected from monkey ICSI and SCNT blastocysts. Essentially, the zona pellucida was first removed from the blastocysts using acid Tyrode's solution. Subsequently, the trophoblasts were isolated from the intact blastocysts with the help of laser assisted cutting.

## RNA-seq library construction

Blastocysts and trophoblasts of ICSI and SCNT groups were separately lysed in lysis buffer containing RNase inhibitor, and cDNA synthesis using SMART-Seq v4 Ultra Low Input RNA Kit for sequencing (Clontech, 634888) according to the user manual. After amplification, the sequencing libraries were made with the fragmented cDNA using TruePrep DNA Library Prep Kit V2 for Illumina according to manufacturer's instructions (Vazyme, TD503). Libraries were quantified using a Qubit dsDNA HS Assay Kit (Thermo Scientific, Q32854), quality-controlled using a Fragment Analyzer instrument (Agilent), and paired-end 150 bp sequencing was performed on a Novaseq 6000 sequencer (Illumina).

## RNA-seq data analysis

The raw sequencing reads were trimmed by fastp or trim_galore with default parameters to remove low-quality bases and adapters in paired-end reads. The remaining reads were mapped to the *Macaca fascicularis* genome (macFas5) with Hisat2[56] with parameters "--no-mixed", "--no-discordant" and "--no-softclip". The SAM files was converted to BAM files using SAMtools[57], then the BAM file were sorted and the index files were created. Raw read counts were calculated by featureCounts[58]. The indexed BAM files were converted to BW files using bamCoverage of deeptools[59]. Expression levels of each gene was quantified with normalized FPKM (Fragments Per Kilobase of exon model per Million mapped fragments). Differential gene expression analysis was performed using the DESeq2 package in the Bioconductor R program (version 4.2.1), two-fold changes, baseMean >10 and $p$-value < 0.01 were used as the cutoff. The Pearson correlation coefficient of gene expression level was calculated using the *cor* function in R to indicate the correlation between replications.

Allele-specific alignment was performed using the masked parental genome with bowtie2. Then, the allelic origin of reads that cover known SNP positions were identified by SNPsplit[60]. The BW files containing expression peaks of the paternal and maternal genomes were generated as described before.

## WGBS library construction for monkey blastocysts, STB cells and trophoblasts

The embryo number used for blastocysts and trophoblasts WGBS library construction was demonstrated as Supplementary Table 1. The library construction of blastocysts, post-implanted embryos (E/A/Y/T tissues), STB cells and trophoblasts were performed according to a previously published protocol[61–63]. Briefly, ICSI, SCNT blastocysts or STB cells were seeded into lysis buffer by mouth pipette and subjected to bisulfite conversion using the EZ DNA Methylation Direct Kit (Zymo Research, D5020). A small amount of unmethylated Lambda DNA (Promega, D152A) was added to each sample before bisulfite conversion to serve as spike-in controls for evaluating bisulfite conversion efficiency. After column-based purification, DNA was complemented with the random primer Preamp and Klenow polymerase (Enzymatics, P7010-HC-L). This random priming was repeated five times in total. Second strands were synthesized using another random primer, Adapter 2. Final libraries were generated after PCR amplification with the Illumina universal PCR primer and Illumina indexed primer, and were then purified using SPRI beads. Final libraries were subjected to paired-end 150 bp sequencing on a NovaSeq 6000 sequencer (Illumina) with PhiX spike-in control.

## Single cell WGBSlibrary construction for oocytes

For the WGBS library construction of oocyte, a single oocyte was collected for each replication. A total of 2 GV and 3 MII oocytes were collected in this research. After cells were lysed using cell lysis buffer, bisulfite treatment was performed and the resultant DNA was purified using PureLink PCR Micro Kit (Invitrogen™, K310050). The purified DNA was added with oligo1, followed by PCR. Then, the single strand DNA template and oligo were digested, and the PCR products was purified using DynabeadsM-280 Streptavidin. The beads were resuspended in reaction mix with oligo 2, the products were then purified again using Agencourt RNAClean XP beads. The purified DNA products were amplified using primers containing Illumina specific primer and index. Finally, the amplified PCR products were further purified and eluted in elution buffer. The library was qualified using Agilent 2100 Bioanalyzer and ABI StepOnePlus Real-Time PCR System (TaqMan Prob). The qualified DNA library was then sequenced under the DNBseq platform.

## Placenta WGBS library construction

For the WGBS of placenta, 1 μg genomic DNA was fragmented to 200–300 bp and purified using MiniElute PCR Purification Kit (QIAGEN). The resulted DNA was incubated at 20 °C with End Repair Mix and then purified. After end repair, the library was combined with A-Tailing Mix and incubated at 37 °C and then purified. Next, methylated adapters were ligated to the Adenylate 3' Ends DNA and purified again. The above library was then undergoing bisulfite treatment and purified with Methylation-Gold kit (ZYMO). After that, fragments ranging from 320 to 420 bp were purified using agarose gel electrophoresis. The library was then amplified using PCR. Finally, the purified DNA products was qualified using the Agilent Technologies 2100 bioanalyzer and ABI StepOnePlus Real-Time PCR system. The library was sequenced in pair-end on the DNBseq platform.

### Syncytiotrophoblast cell isolation from placenta

To isolate syncytiotrophoblast cells (STB), placental tissues were carefully cut into small pieces of approximately $1\,mm^2$ using scissors. The cut tissue fragments were then subjected to digestion using a combination of 0.125% trypsin, 0.05% collagenase IV, and 0.04% DNase I. This digestion process was repeated twice to ensure thorough cell shedding of the tissues. The resulting digested product was diluted with DMEM medium and incubated on a shaking incubator at 37 °C for approximately 8 min. To halt the enzymatic activity, 5% FBS was added to the mixture. The suspension was then filtered through a 70 μm cell filter membrane to remove any debris. The cell suspension was subsequently centrifuged at 4 °C and 300 g for 10 min, allowing for the formation of cell pellet. This cell pellet was gently resuspended in 1 mL DMEM medium containing 5% FBS. The resuspended cell suspension was carefully layered onto a pre-prepared Percoll solution with a 7-layer gradient ranging from 10% to 70% (10%, 20%, 30%, 40%, 50%, 60%, and 70%). The layered suspension was then centrifuged at 1200 g for 15 min. Following centrifugation, the cell layers at 30%, 40%, and 50% gradients were selectively collected. Syncytiotrophoblast (STB) cells were meticulously selected and collected using a micropipette under a stereomicroscope, employing their size as a criterion. To validate the collection of STB cells, a subset of the collected cells was stained with DAPI for nuclear staining. These stained cells were subsequently observed under a microscope to confirm their density as STB cells.

### WGBS data analysis

The raw sequencing reads were trimmed by fastp (version 0.23.2) or trim_galore with default parameters to remove low-quality bases and adapters in paired-end reads. For blastocyst, post-implanted embryos (E/A/Y/T), STB cells, oocytes and trophoblast WGBS data analysis, the remaining reads were aligned to the *Macaca fascicularis* genome (macFas5) using Bismark (version 0.23.1) with parameters "--non_directional[64]". The WGBS datasets of fibroblast, sperm and placenta were aligned to macFas5 using Bismark with default parameters. The methylation level and coverage depth of each cytosine were extracted from the aligned reads with bismark_methylation_extractor. The bedGraph files resulted from bismark_methylation_extractor was converted to BW files using bedGraphToBigWig software. HyperDMRs and hypoDMRs were identified using methpipe and were further filtered requiring at least 25 CpG sites and at least 25% methylation difference[65]. Tiling window/region DNA methylation level was calculated by using methylKit[66] package in the Bioconductor R program. The customed annotation of DMRs was performed using annotatePeaks.pl script of HOMER (Hypergeometric Optimization of Motif EnRichment)[67]. The GTF file containing known genomic regions used for annotation was downloaded from UCSC Genome Browser.

### Sequencing data visualization

DNA methylation peaks of the WGBS dataset and gene expression levels of the RNA-seq dataset were visualized using the UCSC Integrative Genomics Viewer genome browser.

### Primers

The primers used in this study are listed as in the Supplementary Table 8.

### Software and packages

The software and packages used in the study for data processing are listed in the Supplementary Table 9.

### Statistics and reproducibility

No statistical method was used to predetermine sample size. No data were excluded from the analyses. The experiments were not randomized. The investigators were not blinded to allocating during experiments and outcome assessment.

### Reporting summary

Further information on research design is available in the Nature Portfolio Reporting Summary linked to this article.

## Data availability

The RNA-seq datasets of hybrid blastocyst and trophoblasts generated in this study have been deposited in the Gene Expression Omnibus (GEO) database under accession code GSE221634 and GSE239741. The WGBS datasets of hybrid blastocyst and trophoblasts generated in this study have been deposited in the GEO database under accession code GSE221636 and GSE239742. The WGBS dataset of hybrid fibroblast generated in this study has been deposited in the GEO database under accession code GSE221637. The WGBS datasets for placenta, oocyte, STB cells and in vitro cultured post-implanted monkey embryos generated in this study have been deposited in the GEO database under the accession code GSE222930. The whole genome sequencing datasets of parental genomes generated in this study have been directly deposited in the Sequence Read Archive (SRA) under the BioProject number of PRJNA915580. The public WGBS data of human sperm used in this study is available in the GEO database under accession code GSE109344. The raw data used for statistical graphs in this study is available in the Source Data folder at Figshare [https://doi.org/10.6084/m9.figshare.23957904].

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

## Acknowledgements

We thank Dr. Mu-ming Poo for his comments on the manuscript, members of our laboratories for their contributions, and the staff of the Non-human Primate Facility of the Center for Excellence in Brain Science and Intelligence Technology for their assistance in animal care. This work was supported by grants from the National Natural Science Foundation of China Grant (31825018, 82021001 to Q.S.), the National Key Research and Development Program of China (2022YFF0710901 to Q.S., 2018YFA0107001 and 2020YFA0804000 to F.L.), the Strategic Priority Research Program of the Chinese Academy of Sciences (XDB32060100 to Q.S. and Z.L.), the Shanghai Municipal Science and Technology Major Project (2018SHZDZX05 to Q.S. and Z.L.), the National Science and Technology Innovation 2030 Major Program 2021ZD0200900 and Lingang Lab (Grant LG202106-01-01 to Q.S. and LG202106-02-01 to Y.L.), the From 0 to 1 Original Innovation Project of the Basic Frontier Scientific Research Program of the Chinese Academy of Sciences (ZDBS-LY-SM019 to Z.L.).

## Author contributions

Q.S., Z.L. and F.L. conceptualized and supervised the project. Z.L. and Z.D.L. designed the experiments. Z.D.L. performed ICSI, SCNT, TR, monkey embryo electrofusion, primary fibroblast isolation, embryo transfer and AR staining. Z.D.L. conducted the chimerism examination and STR analysis. J.X.Z., Z.Y.N. and J.W.L. constructed the RNA-seq and WGBS libraries. Z.D.L., S.Y.S. and J.C. performed donor cell cultivation. Y.Z.L. and Z.D.L. performed in vitro transcription. Z.D.L., Y.Z.L., and S.Y.S. performed mtDNA genome sequencing and analysis. Z.D.L., J.X.Z, Z.L. and S.Y.S. integrated and interpreted the data. Y.T.X., C.Y.L. and Y.H.N. collected the oocytes and sperm. Q.S., Z.L. and F.L.L. provided conceptual advice and discussed results. Z.D.L. wrote the manuscript with the input from other authors.

## Competing interests

The authors declare no competing interests.
