## [Peer Review File · Nature Communications]

REVIEWER COMMENTS

Reviewer #1 (Remarks to the Author):

In this study, Liao et al. attempted to address the issue of low successful rate of monkey cloning. By comparing SCNT and ICSI blastocysts, the authors found a global decrease in DNA methylation and loss of imprinting of the maternal imprinted genes in the SCNT embryos. Using SNP information, they identified 4 genes whose dysregulation correlates with their changes in imprinting state. Histological analysis revealed placenta hyperplasia and calcification in SCNT embryos which prompted them to test whether they can use trophoblast replacement (TR) to rescue the placenta defects to achieve monkey cloning, which allowed them to successfully cloned one rhesus monkey that survived for 2 years. They also showed that the TR cloned monkey placenta restored the imprinting of the 4 genes with imprinting defects in SCNT. Thus, the imprinting defects correlate with the cloning failure.

In general, this study is well performed and the authors have put into a lot of work. Although only one live animal is obtained, it is the first rhesus monkey cloned that survived for 2 years. Despite it is not proving (due to only a single monkey, which could be a random event), the authors provided correlative data indicating that TR-SCNT corrected some imprinting defects in SCNT, which may contribute to the generation of live monkey. That being said, it is not clear whether correction of the 4 imprinted genes actually contributed to the survival of the cloned monkey. It is also not clear how many other imprinting are also defective in the SCNT blastosysts as the number of imprinted genes the author examined are very limited due to the available SNP information. The authors can improve their work by addressing the following questions:

1. The authors identified 420 differentially expressed genes by comparing SCNT and SCNT-TE blastocysts. Of the 420 DEGs, they only identified 4 imprinted genes using SNP information. The authors can expand their analysis by using the SNP-free method (PMID: 35310079). They might be able to identify more genes with imprinting defects, thus can better link the imprinting defect to cloned monkey survival. This way, they might be able to provide a genome-wide view of the imprinting changes ICSI, SCNT and SCNT-TR placenta, which is better than their current presentation in Fig. 5f and S4.
2. Data shown in Fig. S5g and S5h indicate that the thickness of SCNT-TR and SCNT placenta is comparable, and are both significantly thicker than ICSI placenta. However, in Supplementary Table 3, the average thickness of SCNT-TR placenta is only 8.85 mm, which is less than the 10.01 mm of ICSI placenta. The two pieces of data are conflict. The bar graph of #14 on Fig. S5g does not match the data in table S3.
3. In Supplementary Table 4 and 5, the authors used transferred embryos to calculate live birth and survival rate. However, they used total activated embryos instead of total transferred embryos to calculate for SCNT-TR in Table 1. Could the author explain why they used different ways to calculate the developmental efficiency?

4. Can the authors provide more quantitative data when analyzing allelic DNA methylation and gene expression instead of just presenting snapshots (in Figure 4 and Figure S2)?

Reviewer #2 (Remarks to the Author):

The paper by Liao et al. reports the first birth of cloned rhesus monkey following trophoblast replacement (TR). They also performed comprehensive DNA methylation and transcriptome analysis of cloned preimplantation embryos and postimplantation embryos/placentas. By combining the WGBS and RNA-seq data, they found the inherited loss of maternal imprinting for several imprinted genes in the pre- and post-implanted monkey SCNT embryos, consistent with a previous study (ref. 39). Importantly, the four identified genes were also imprinted or bias-expressed genes in humans. Furthermore, their histological analysis of cloned placentas revealed excessive calcification and enlarged thickness of the placental tissue. The authors finally applied the similar sets of epigenetic/histological analyses to cloned newborn's placenta and corrections of clone-associated abnormalities except for the thickness of the placenta. Overall, the manuscript is well-written, the analytical data seem to be reliable with enough number of samples, and the all the results were obtained with a high skill of handling monkey embryos in a large monkey facility. However, the main drawback of this paper is the uncertainty as to whether TR was actually effective in clone's birth. With only one born, it is scientifically impossible to link TR to clonal births. The authors discussed about the possible significance of the corrected expressions of aberrantly expressed imprinted genes, but this would not be sufficient to convince the readers. Of course, the best solution would be to obtain a second clone, but we understand that it is too much to ask for that in the revision of this paper. However, the significance of this paper in the present form, both biologically and practically, is halfway.

One way to increase the significance of this paper is to include a genetic analysis. Rhesus monkey clones are known to have unstable chromosome behavior (Simerly et al. Science 2003; PMID12690191). Many of the embryos produced by the authors might have had chromosomal abnormalities. If so, it is not promising to do further epigenetic analysis in this way to resolve the aberrations. Also, if there are genetic abnormalities, it will affect the results of the epigenetic analysis, too. Whole genome analysis is not necessary, and analysis of the constitution of the M-phase chromosomes in cloned embryos and/or chromosomes behavior analysis immediately after nuclear transfer would be sufficient.

Minor comments:

1. In the discussion, the authors mentioned the putative functions of aberrantly expressed imprinted genes. It may be important to further discuss the possibility of whether these putative functions might be related to the SCNT-specific developmental arrest at around day 40 of gestation.
2. The authors identified several genes that were associated with DNA methylation changes in cloned embryos. Were these DNA methylation changes also found in donor somatic cells? i.e, carry over from the donor cells? If not, why did these DNA methylation changes occur in cloned embryos?

3. Line 340: (Fig. "5b" and 5c right)

4. Extended Data Fig. 6e: Both Model 1 and Model 2 seem to form 4-cell embryos. Is this true? In the usual developmental course, 4n fused embryos (1-cell like) cleave into 2-cells. Indeed, the Model 1 embryo at metaphase contains only one metaphase chromosomal mass. It is impossible for it to divide into four blastomeres at a time.

Reviewer #3 (Remarks to the Author):

Reprogramming mechanism dissection and trophoblast replacement application in monkey somatic cell nuclear transfer by Zhaodi Liao et al. (NCOMMS-23-11496-T)

In this manuscript authors report on their efforts to produce live rhesus macaque offspring by SCNT using fetal fibroblasts. In short, the authors produced and transferred 484 rhesus SCNT blastocysts into recipients. While 35 implantations were recorded, most pregnancies were lost and only one progressed to live birth but infant died 23 hours after birth. Authors then conducted RNA-seq, and whole genome bisulfite sequencing analyses of control IVF and SCNT blastocysts that showed DNA methylation and imprinted gene expression differences. These differences were also observed during fetal development. Morphological defects were also observed in placenta of SCNT pregnancies.

Assuming that extraembryonic/placental defects are primary cause of SCNT fetal losses, authors first attempted to develop rhesus tetraploid embryos for trophoblast complementation. However, some transferred tetraploid embryos produced diploid live born offspring indicating spontaneous ploidy correction. Finally, authors produced trophoblast vesicles by surgically removing the ICM from IVF blastocysts and transplanting into these vesicles ICMs isolated from SCNT blastocysts. Transfer of 11 such reconstructed SCNT blastocysts produced one live born monkey that survived for over 2 years.

Overall, the results presented in this study are based on massive number of rhesus SCNT embryos analyzed or transferred to recipients. Authors also provided thorough analysis of embryonic, fetal and postnatal survival of SCNT embryos. Moreover, full term development of tetraploid and ICM deficient rhesus macaque IVF blastocysts were also studied.

Major comments:

1. Authors claim that trophoblast replacement described in this manuscript rescued postnatal development of rhesus monkey SCNT blastocysts. However, this conclusion is based on just one infant produced from each treatment group and thus cannot be statistically valid. Nevertheless, the technique

of trophoblast/trophectoderm replacement in blastocysts is novel and authors should highlight this advance and discuss potential applications for conventional IVF to rescue development of fertilized embryos with defects in trophectoderm.

2. Results on the transcriptional and DNA methylation profiling lack details on number of SCNT embryos and replications used for each experiment. It is well known that quality of SCNT embryos is highly heterogeneous, thus each individual SCNT blastocyst could display its own unique set of abnormalities. Authors should present data showing range of transcriptional and epigenetic defects within the SCNT group.

3. It would be more informative if transcriptional and DNA methylation profiling were done on trophectoderm vesicles rather than on whole blastocysts, since authors develop this approach. This comparison would clearly define defects in the extraembryonic lineages.

Dear Reviewers,

Reviewer 1#:

Thank you for your thorough evaluation and constructive feedback on our study, we highly appreciate your input.

We acknowledge that obtaining only one live animal is a limitation of our study. We agree that our data only provides correlative evidence suggesting that SCNT-TR corrected certain imprinting defects in SCNT, potentially contributing to the generation of a live monkey. Further studies in the future are necessary to determine the extent to which the correction of these genes actually contributed to the survival of the cloned monkey. Additionally, we acknowledge that the limited number of imprinted genes we examined is a drawback of our study, which was attributed to the limited SNP information available. However, we believe that the successful cloning of a rhesus monkey that survived normally for a duration of 2 years signifies a notable progression in the field.

Here, we would like to address your questions as follows:

1. The authors identified 420 differentially expressed genes by comparing SCNT and SCNT-TE blastocysts. Of the 420 DEGs, they only identified 4 imprinted genes using SNP information. The authors can expand their analysis by using the SNP-free method (PMID: 35310079). They might be able to identify more genes with imprinting defects, thus can better link the imprinting defect to cloned monkey survival. This way, they might be able to provide a genome-wide view of the imprinting changes ICSI, SCNT and SCNT-TR placenta, which is better than their current presentation in Fig. 5f and S4.

We highly value your suggestion to expand our analysis by utilizing the SNP-free method (PMID: 34619096 and 35310079) to identify additional genes exhibiting imprinting defects in cloned embryos and to provide a comprehensive overview of the changes in imprinting.

In the mentioned paper (PMID: 34619096), there are two SNP-free methods: TARSII (tissue-associated, reads-based, SNP-free method for identifying imprint DMRs) and CARSII (CpG-island-associated, reads-based, SNP-free method for identifying imprint DMRs). However, the application of the TARSII method is not feasible in our research since it requires the integration of methylomes from multiple tissues. On the other hand, with the CARSII method, the authors were able to confine the identification of candidate maternal germline DMRs (mgDMRs) to emDMCs (early-embryonic, maternal-allele-methylated differential methylated CpG-islands) within a single somatic tissue. To implement CARSII, the authors initially identified emDMCs utilizing the methylomes of early embryos or germline cells. Subsequently, they assessed the enrichment of both hyper/hypomethylated reads on these emDMCs in every somatic tissue and excluded unsuitable candidates. The remaining mgDMR candidates were identified as potential maternal germline differentially methylated CpG islands (mgDMCs) in each somatic tissue.

Using 2,657 emDMCs identified in the sperm and oocyte methylomes of *Macaca mulatta*, researchers (PMID: 34619096) discovered 131 maternally-biased mgDMRs (maternal germline differentially methylated regions) in monkey ICSI placentas and 45 in SCNT placentas. This finding suggests the loss of maternal-biased DNA methylation in monkey SCNT placenta.

Following the reviewer's suggestion, we have employed the CARSII method to identify tissue-specific mgDMRs or mgDMCs in monkey blastocyst and placenta samples.

To conduct the CARSII analysis, we utilized a *bed* file containing all CpG islands, a sorted *sam* file with duplicates removed by Picard, and a *wig* file representing DNA methylation levels at base resolution.

For this purpose, we adopted the *Macaca fascicularis* (*M.f*) and *Macaca mulatta* (*M.m*) emDMCs data from the paper (PMID: 34619096), which revealed 2,657 emDMCs identified between haploid androgenic (AG) and parthenogenetic (PG) 16-cell embryos of *M.f*, as well as 2,321 emDMCs identified between the gametes of *M.m*.

The *bam* files for each sample were generated using Bismark and deduplicated using the Picard tools. These resulting *bam* files were then converted into *sam* format using Samtools.

Additionally, the *wig* files containing the DNA methylation levels at base resolution of each sample was generate using Bismark.

Following the instructions from the STAR Protocol (PMID: 35310079), we inputted these required files (*bed*, *sam*, *wig*) into the CARSII software, ensuring that all commands strictly adhered to the provided guidelines.

Initially, we utilized the *bed* file containing 2,657 *M.f* emDMCs as the input for the analysis. However, we did not identify any putative DMRs in the monkey ICSI blastocysts, and only 4 putative DMRs in the monkey SCNT blastocysts. When we used the *M.m* emDMCs as input, we observed zero putative DMR in the monkey ICSI, but 36 and 51 potential DMRs in the two replicates of SCNT blastocysts (SCNT1 and SCNT2), respectively. These results contradict the previous report (PMID: 34619096).

Moving on to the placentas, we found no mgDMRs in any of the placenta methylomes when using the *M.f* emDMC.bed as the input bed file. When using the *M.m* emDMC.bed as the input bed file, we identified no mgDMRs in 2 out of 3 ICSI placentas, while only 2 DMRs were found in the remaining ICSI placenta. In the cloned monkey placenta, we discovered 4, 3, and 2 DMRs, respectively, while no DMRs were identified in the TR placenta.

Although the CARSII method proved to be a valuable tool for identifying potential imprinting genes without the need for SNP information, as demonstrated by the previous study of imprinting loss in the cloned monkey placenta, our study did not yield meaningful findings regarding imprinting loss using this method. Consequently, we have to abandon this method for identifying imprinting loss in our study.

2. Data shown in Fig. S5g and S5h indicate that the thickness of SCNT-TR and SCNT placenta is comparable, and are both significantly thicker than ICSI placenta. However, in Supplementary Table 3, the average thickness of SCNT-TR placenta is only 8.85 mm, which is less than the 10.01 mm of ICSI placenta. The two pieces of data are conflict. The bar graph of #14 on Fig. S5g does not match the data in table S3.

We apologize for the confusion caused by the inconsistency between the thickness of the SCNT-TR placenta presented in Supplementary Table 3 and the bar graph #14 in Fig. S5g of our original submission (Extended Data Figure 9g of the revised manuscript). We reviewed our original data and found that the average thickness of SCNT-TR placenta should be 17.39 mm instead of 8.85 mm, as presented. We have made the necessary corrections in our revised manuscript and taken measures to ensure the accuracy of our data presentation (as highlighted in line 392 of the revised manuscript, Supplementary Table 3). We thank the reviewer for finding our mistake in data presentation.

3. In Supplementary Table 4 and 5, the authors used transferred embryos to calculate live birth and survival rate. However, they used total activated embryos instead of total transferred embryos to calculate for SCNT-TR in Table 1. Could the author explain why they used different ways to calculate the developmental efficiency?

We appreciated your comment regarding the calculation of developmental efficiency in Supplementary Tables 4 and 5, as well as Table 1.

In Supplementary Table 4, the embryos were observed for their pre-implant development before being transferred into the surrogate mothers' uterus. Similarly, for the embryos in Supplementary Table 5, they had to reach the blastocyst stage for ICM removal. However, it is important to note that not all the activated or fertilized embryos were utilized for embryo transfer. Therefore, it is difficult to precisely determine the number of fertilized or activated embryos that successfully developed into blastocysts in these Supplementary Tables. As a result, we relied on the transferred embryos for the calculation of the developmental efficiency in these two tables.

On the other hand, none of the 484 "Total embryos activated" SCNT embryos listed in Table 1 were transferred *in vivo* at their blastocyst stage. In fact, nearly all of these 484 activated embryos were transferred into surrogates at the 2-cell stage. In this particular scenario, the count of embryos transferred corresponds to the number of the activated

embryos. Hence, we used the activated embryos for the sake of convenience when comparing SCNT and SCNT-TR embryos in Table 1.

4. Can the authors provide more quantitative data when analyzing allelic DNA methylation and gene expression instead of just presenting snapshots (in Figure 4 and Figure S2)?

We thank the reviewer for this suggestion. In our revised manuscript, we have included quantitative data on *THAP3* and *RBOBTB3*, specifically analyzing their allelic DNA methylation and gene expression. This addition aims to offer a more informative and comprehensive exploration of their changes observed in cloned monkey embryos. (Extended Data Fig. 4d-f, lines 185-190 and lines 199-202 in the revised manuscript)

One thing to be noted that the SNPs at of *RHOBTB3* locus we identified between the parental genomes of ICSI embryos reside in the intron regions but not in the exon regions, which makes us unable to accurately quantify the parental specific expression of *RHOBTB3* in ICSI embryos (Extended Data Fig. 4f in the revised manuscript).

Reviewer 2#:

Thank you for taking your time to read and provide feedback on our work. We acknowledge your concern regarding the uncertainty of whether TR was effective in clone's birth rate, as we only had one successful cloning birth in this study.

We have addressed your minor comments below:

1. One way to increase the significance of this paper is to include a genetic analysis. Rhesus monkey clones are known to have unstable chromosome behavior (Simerly et al. Science 2003; PMID12690191). Many of the embryos produced by the authors might have had chromosomal abnormalities. If so, it is not promising to do further epigenetic analysis in this way to resolve the aberrations. Also, if there are genetic abnormalities, it will affect the results of the epigenetic analysis, too. Whole genome analysis is not necessary, and analysis of the constitution of the M-phase chromosomes in cloned embryos and/or chromosomes behavior analysis immediately after nuclear transfer would be sufficient.

As you pointed out, a previous report has demonstrated that rhesus monkey SCNT embryos can exhibit unstable chromosome behavior, which may impact the efficiency of embryo development (DOI: 10.1126/science.1082091). Therefore, it is good to verify the chromosomal constitution of the cloned embryos and investigate any potential chromosomal abnormalities in cloned embryos generated in our study.

In the revised version of our study, we have included a section focusing on the chromosomal analysis of the cloned embryos to investigate their organization. In order to

accomplish this, we performed immunostaining of the monkey SCNT embryos at the 1-cell stage to examine their first mitotic spindle assembly. The results, as illustrated in Extended Data Figure 1, indicate that the chromosomes in the SCNT monkey embryos exhibited well-aligned and properly positioned microtubules during the first mitosis. Clear bipolar spindles were observed in the monkey SCNT embryos. This outcome could potentially be attributed to the treatment with epigenetic modification factors, such as *Kdm4d* and Tricostatin A, in the monkey SCNT embryos. Consequently, we concluded that chromosome abnormalities did not occur in the monkey SCNT embryos generated in our study.

These observations have been added in the Results section (lines 87-89 and Extended Data Figure 1), and the method for immunostaining has been included in the Methods section (lines 543-563) of our revised manuscript. We thank the reviewer for this suggestion, which have help us to improve the scientific validity and significance of our research in terms of chromosomal behavior in monkey SCNT embryos.

2. In the discussion, the authors mentioned the putative functions of aberrantly expressed imprinted genes. It may be important to further discuss the possibility of whether these putative functions might be related to the SCNT-specific developmental arrest at around day 40 of gestation.

Thank you for bringing up this valuable suggestion. In our revised manuscript, we have discussed the potential involvement of imprinting gene loss in the developmental arrest observed in SCNT monkey fetuses during early implantation stages (lines 444-453).

It would indeed be important to further explore the possibility of whether the loss of imprinting genes could be related to the SCNT-specific developmental arrest around day 40 of gestation. This could shed light on the underlying mechanisms behind the observed developmental abnormalities in SCNT embryos.

3. The authors identified several genes that were associated with DNA methylation changes in cloned embryos. Were these DNA methylation changes also found in donor somatic cells? i.e, carry over from the donor cells? If not, why did these DNA methylation changes occur in cloned embryos?

Thank you for your inquiry regarding the DNA methylation changes observed in the cloned embryos and their potential association with the donor somatic cells.

In our original manuscript, we have provided evidence in Figure 4e and Extended Data Figure 2f-h (Figure 4h and Extended Data Fig. 4a-c in the revised version) to demonstrate that the DNA methylation changes of these genes were indeed inherited from the donor cells. These figures comprehensively compared the DNA methylomes of oocytes, sperm,

fibroblasts, ICSI blastocysts, and SCNT blastocysts. We presented the DNA methylation levels of each gene (*DNMT1*, *THAP3*, *SIAH1* and *RHOBTB3*) in each sample, highlighting the differentially methylated regions (DMRs) with red boxes.

Our findings indicate that the DMRs of these genes maintain their maternal DNA methylation from the oocyte stage to the late blastocyst stage in ICSI embryos. However, we observed a loss of DNA methylation in both fibroblast and SCNT blastocysts, suggesting an inherited loss of DNA methylation from somatic cells to SCNT blastocysts. (lines 203-207 and 409-414 in the revised version)

It is worth mentioning that when we investigated the DNA methylation of the ICSI and SCNT trophoblast, we noticed that replicate one showed complete loss of DNA methylation at the DMR of *THAP3*, *DNMT1*, *RHOBTB3* and *SIAH1*, while replicate two showed partial preservation DNA methylation at these loci (Extended Data Fig. 5g-j in the revised version). This suggested that DNA methylation at these loci in the donor cells for replicate two SCNT sample might have escaped the DNA methylation loss at these loci. (lines 435-443 in the revised version)

We hope that these results could effectively address your concerns regarding the DNA methylation changes observed in these identified genes.

4. Line 340: (Fig. "5b" and 5c right)

We apologize for missing "Figure 5b" in line 340 of the original manuscript. We have now added it in our revised version (line 380 in the revised manuscript).

5. Extended Data Fig. 6e: Both Model 1 and Model 2 seem to form 4-cell embryos. Is this true? In the usual developmental course, 4n fused embryos (1-cell like) cleave into 2-cells. Indeed, the Model 1 embryo at metaphase contains only one metaphase chromosomal mass. It is impossible for it to divide into four blastomeres at a time.

Thank you for your comment on the Model 1 and Model 2 embryos' cleavage patterns (Extended Data Fig. 6e in our original submission). We agree with your opinion that 4N fused embryo won't become a 4-cell embryo in one round of cleavage. We have now changed our description as below (line 310~324 of the revised manuscript):

"To investigate why electrofused monkey embryos were able to develop into live fetuses, we performed live imaging using the electrofused embryos with H2B-EGFP overexpression. Our findings revealed two different types of mitosis models in these embryos (Extended Data Fig. 10e). In model 1, the two nuclei initially come close to each other within the newly formed blastomere after the fusion of cleavage cells, and then divide into four new blastomeres, with each blastomere containing one nucleus. In this model, the two nuclei approach each other before undergoing the first mitosis. However, further investigation is needed to determine whether these two nuclei undergo fusion to create a new, single

nucleus or if they remain separate but closely juxtaposed prior to the M-phase. In Model II, despite the fusion of two cleavage cells and the formation of a new blastomere, the two nuclei do not come close to each other but independently undergo the first mitosis within the single fused blastomere, resulting in the formation of four blastomeres. These observations suggested the existence of a unique phenomenon in maintaining chromosomal ploidy stability in primate embryos, the specific mechanism of which is awaiting further exploration and verification."

We hope our updated description fulfill your concerns regarding the division pattern of Model 1 embryos.

Reviewer 3#:

Thank you for your thoughtful review of our manuscript. We are glad to hear that you found our experimental approach and analyses to be thorough and comprehensive.

We have addressed each of your comments below:

1. Authors claim that trophoblast replacement described in this manuscript rescued postnatal development of rhesus monkey SCNT blastocysts. However, this conclusion is based on just one infant produced from each treatment group and thus cannot be statistically valid. Nevertheless, the technique of trophoblast/trophectoderm replacement in blastocysts is novel and authors should highlight this advance and discuss potential applications for conventional IVF to rescue development of fertilized embryos with defects in trophectoderm.

We acknowledge that our conclusion regarding trophoblast replacement rescuing postnatal development of rhesus monkey SCNT blastocysts is based on a small number of individuals. Therefore, we have now shortly discussed the potential applications of this technique for conventional IVF to rescue development of fertilized embryos with defects in trophectoderm in the Discussion section of the revised manuscript as suggested (lines 469-486).

2. Results on the transcriptional and DNA methylation profiling lack details on number of SCNT embryos and replications used for each experiment. It is well known that quality of SCNT embryos is highly heterogeneous, thus each individual SCNT blastocyst could display its own unique set of abnormalities. Authors should present data showing range of transcriptional and epigenetic defects within the SCNT group.

We acknowledge that the quality and characteristics of SCNT embryos can vary extensively, leading to inherent heterogeneity within the SCNT group.

We have included the necessary information regarding the number of SCNT embryos used and the number of replications performed for each experiment in Supplementary Table 1 of both our original and revised manuscripts. This table presents an overview of the number of embryos used for each replication and the number of replicates for each dataset. This table also contain the parental information of the embryos used in this study.

3. It would be more informative if transcriptional and DNA methylation profiling were done on trophoctoderm vesicles rather than on whole blastocysts, since authors develop this approach. This comparison would clearly define defects in the extraembryonic lineages.

We agree that conducting transcriptional and DNA methylation profiling on trophoctoderm vesicles, rather than whole blastocysts, would provide more information and enable better identification of defects in the extraembryonic lineages.

In response to this suggestion, we have added the transcriptome and DNA methylation analyses of SCNT and ICSI trophoctoderm in the revised manuscript. Although the analysis on trophoctoderm vesicles provided limited information due to limited SNP information in the samples, we did observe that two of the imprinted genes, which exhibited biallelic expression in blastocysts, became biallelically expressed in the SCNT trophoblasts (Extended Data Fig. 5, lines 310-324 and lines 435-443 of the revised manuscript).

Once again, we greatly appreciate the insightful comments and suggestions from the three reviewers, as they have been instrumental in enhancing the scientific rigor and clarity of our work. We sincerely thank you for the careful review and for ensuring that all essential information is included in our manuscript. If you have any further suggestions or concerns, please do not hesitate to inform us.

REVIEWERS' COMMENTS

Reviewer #1 (Remarks to the Author):

The authors tried to address my comments. While their response to my comments #2 and #4 are reasonable, their response to my comments #1 and #3 are less satisfying.

For my comment 1, the authors claimed to have followed the suggested CARSII method but did not identify new mgDMRs in their placenta samples which is surprising. Based on the previous description, the CARSII method performs best in somatic tissues with global DNA methylation above 80%. Thus, it is possible that the global placenta methylation in this study is much lower than that in the study of the original CARSII method. The same issue appears also exist in the mgDMR prediction in the blastocyst samples of this study where the global DNA methylation is much lower than expected. While it may take some effort, I strongly suggest that the authors try to adjust the ratio of hypermethylated reads in mgDMRs (probably cut down a little from 0.3, eg. 0.25) and redo the analysis to see whether they can increase the coverage.

For comment 3, according to the authors, in Table 1 they transferred 2-cell embryos for SCNT-WT while transferred the blastocyst for SCNT-TR due to the additional reconstruction step in SCNT-TR. If the same culturing conditions are used, reconstructed SCNT-TR blastocysts are expected to have an increased chance of death compared to that of SCNT-WT blastocysts. Thus, the calculation and comparison of the implantation rate, live birth and survival rate between SCNT-WT and SCNT-TR may be problematic, because they ignored the differences of blastocyst survival rate between SCNT-WT and SCNT-TR. If there is no better way to address this issue, the authors should add a discussion to highlight this issue clearly.

Reviewer #2 (Remarks to the Author):

All the concerns have been adequately addressed in this revised version, although it was unexpected that the mitotic chromosomes in all (n=13) SCNT embryos were normal (Extended Data Figure 1) because even normally fertilized embryos may occasionally show aberrant chromosomes in humans and monkeys. I have no further comments on this manuscript.

Reviewer #3 (Remarks to the Author):

My comments were addressed in the revised manuscript.

Dear reviewer #1:

We extend our gratitude for your thoughtful review of our manuscript. We have carefully considered your comments and would like to address your concerns as the following.

Comment 1: We appreciate your concern regarding the mgDMR analysis and the importance of achieving comprehensive coverage for reliable results. We have indeed adjusted the ratio of hypermethylated reads in mgDMRs as you suggested, lowering it to 0.25, and reran the analysis. When utilizing the emDMC regions of *Macaca fascicularis*, our results revealed 97 and 94 candidate DMCs in SCNT blastocysts (replicate 1 and replicate 2), respectively. Employing the emDMC regions of *Macaca mulatta*, we identified 10 representative DMCs in replication 1 and replication 2 of monkey SCNT blastocysts, respectively. However, none of the representative DMCs were found in replication 1 and replication 2 of monkey ICSI blastocysts, whether using *Macaca fascicularis* or *Macaca mulatta* emDMC regions. Regarding the placenta, under the 0.25 threshold for hypermethylated read ratio, only 2 DMCs were detected in one of the ICSI placentas, while 1 and 8 DMCs were observed in two of the SCNT placentas.

Comment 3: Your observation regarding the calculation and comparison of implantation rates and live birth outcomes between SCNT-WT and SCNT-TR embryos is duly noted. We actually have already addressed this concern by basing our comparisons on embryos activated rather than utilizing later-stage embryos (line 581~582 of our revised manuscript). However, we appreciate your advice, and we have provided a more explicit discussion highlighting our approach of comparisons (line 583~587 of our revised manuscript). We believe this will ensure that readers are fully informed about the key factors influencing the interpretation of the data.

Once again, we sincerely thank you for your diligence and valuable feedback. Your input undoubtedly helped us improve the quality and rigor of our study.

Dear Reviewer #2,

We appreciate your positive feedback and your acknowledgment that all the concerns you raised have been adequately addressed in the revised version of our manuscript.

We acknowledge your observation regarding the normal mitotic chromosomes in SCNT embryos. In our examination, we utilized 13 and 19 monkey ICSI and SCNT embryos, respectively, to study the first mitosis post-fertilization or activation. It is worth noted that the mislocated MTOC in our ICSI embryo can be observed in Supplementary Figure 1. It is also indeed accurate that primate ICSI embryos sometimes show aberrant chromosomes, accompanied by cell debris surrounding the two blastomeres of 2-cell stage embryos. However, in case of monkey SCNT embryos in this study and our previous reports, the SCNT embryos exhibit a flawless two-cell appearance without any accompanying cell debris. This consistency aligns with the observation of the pristine spindle-chromosome organization during their first mitosis, as depicted in Supplementary Figure 1.

Thank you once again for your contributions to the review process.

Dear Reviewer #3,

Thank you for your feedback, and we are pleased to hear that your comments have been addressed in the revised manuscript.

Your expertise and insights have been invaluable to our research, and we are grateful for your time and effort in reviewing our manuscript. Your contributions to the review process are highly appreciated.